# On the Adversarial Robustness of Multi-Kernel Clustering

Hao Yu [1]   Weixuan Liang [1]   Ke Liang [1]   Suyuan Liu [1]   Meng Liu [1]   Xinwang Liu [1]

## Abstract

Multi-kernel clustering (MKC) has emerged as a powerful method for capturing diverse data patterns, offering robust and generalized representations of data structures. However, the increasing deployment of MKC in real-world applications raises concerns about its vulnerability to adversarial perturbations. While adversarial robustness has been extensively studied in other domains, its impact on MKC remains largely unexplored. In this paper, we address the challenge of assessing the adversarial robustness of MKC methods in a black-box setting. Specifically, we propose *AdvMKC*, a novel reinforcement-learning-based adversarial attack framework designed to inject imperceptible perturbations into data and mislead MKC methods. AdvMKC leverages proximal policy optimization with an advantage function to overcome the instability of clustering results during optimization. Additionally, it introduces a generator-clusterer framework, where a generator produces adversarial perturbations, and a clusterer approximates MKC behavior, significantly reducing computational overhead. We provide theoretical insights into the impact of adversarial perturbations on MKC and validate these findings through experiments. Evaluations across seven datasets and eleven MKC methods (seven traditional and four robust) demonstrate AdvMKC's effectiveness, robustness, and transferability.

## 1. Introduction

Multi-kernel clustering (MKC) leverages multiple kernel functions to capture diverse data patterns (Liang et al., 2024a; Yu et al., 2024; Sun et al., 2024c). By integrating various kernels, MKC addresses the limitations of single-kernel methods, offering a more comprehensive and robust

representation of data structures (Zhang et al., 2023). This enhanced representation improves clustering robustness and generalization, particularly in heterogeneous scenarios. As a result, MKC has been widely applied in domains such as disease diagnosis (Lu et al., 2020; Jiang et al., 2023) and hyperspectral image classification (Jiang et al., 2024; Sima et al., 2022), effectively processing intricate data.

Despite its advantages, the increasing adoption of MKC raises concerns about its robustness against adversarial perturbations (i.e., small, imperceptible modifications) that can significantly degrade performance (Szegedy et al., 2014; Goodfellow et al., 2015; Sun et al., 2024a). Such vulnerabilities, demonstrated in domains like image classification (Shen et al., 2021) and natural language processing (Alzantot et al., 2018), pose significant risks to security-critical applications. This highlights the urgent need to evaluate the adversarial robustness of MKC methods.

However, this area remains largely underexplored. Existing studies primarily address noise with predefined distributions (e.g., Gaussian noise (Sun et al., 2024b; Wang et al., 2023; Yu et al., 2025)) or internal issues such as missing or corrupted data (Liu et al., 2020), while neglecting adversarial attacks involving intentional perturbations (Yang et al., 2023). Furthermore, current research on adversarial robustness in multi-view clustering is limited to deep multi-view clustering methods (Huang et al., 2024), which rely on deep representations rather than kernel-induced similarity matrices central to MKC. This gap prompts a critical question: *Can adversarial attacks be effectively designed to evaluate the robustness of MKC methods?* Addressing this question introduces two main challenges: 1) Designing a learning paradigm that manages the instability of clustering results during adversarial perturbation optimization, especially given the unsupervised nature of clustering and the instability of $K$-Means-based methods (Ben-David et al., 2007). 2) Reducing the computational overhead associated with clustering operations, which are inherently expensive.

**Our Work**. In this paper, we address the problem of adversarial attacks on MKC by formulating it as a reinforcement learning (RL) task. In this setting, adversarial perturbations are injected into the data, and the degradation in clustering performance serves as a reward to guide the optimization process. To tackle these challenges, we propose *AdvMKC*,

---

[1]College of Computer Science and Technology, National University of Defence Technology, Changsha, China. Correspondence to: Xinwang Liu <xinwangliu@nudt.edu>.

*Proceedings of the $42^{nd}$ International Conference on Machine Learning*, Vancouver, Canada. PMLR 267, 2025. Copyright 2025 by the author(s).

a novel black-box adversarial attack framework specifically designed for MKC. The framework leverages RL techniques, particularly proximal policy optimization (Schulman et al., 2017), to address the inefficiency caused by unstable clustering results by employing the advantage function (Pan et al., 2022; Babaeizadeh et al., 2017) during optimization. To further enhance efficiency and reduce computational costs, AdvMKC introduces a *generator-clusterer* framework. In this setup, the generator produces adversarial perturbations, while the clusterer approximates the behavior of MKC methods, significantly minimizing the need for repeated clustering operations during the attack process. Our main contributions are summarized as follows:

- To the best of our knowledge, this is the first study to evaluate the adversarial robustness of MKC methods in the *black-box* setting. We assume an attacker injects imperceptible, intentional perturbations to mislead unknown MKC methods, leading to erroneous clustering results.

- We propose AdvMKC, a novel reinforcement-learning-based adversarial attack framework consisting of a perturbation generator and a clusterer. The generator creates adversarial perturbations, while the clusterer mimics MKC behavior to accelerate optimization.

- We provide theoretical insights into the impact of adversarial perturbations on MKC performance and validate these findings through empirical evaluations.

- We conduct extensive experiments on seven benchmark datasets and eleven MKC methods (seven traditional and four robust methods) to demonstrate the effectiveness, robustness, and transferability of AdvMKC.

## 2. Background and Related Work

### 2.1. Multi-Kernel Clustering

MKC enhances clustering performance by overcoming the limitations of traditional methods such as $K$-Means (Mac-Queen et al., 1967) and spectral clustering (Ng et al., 2001), which rely on single similarity measures and struggle with the complexity of multi-view datasets (Yang et al., 2024; Liang et al., 2024a; Liu et al., 2022a;b). By integrating multiple kernels, each capturing distinct data characteristics, MKC achieves improved accuracy and robustness.

For single-view data $\mathbf{X} \in \mathbb{R}^{N \times d}$, where $N$ is the number of samples, $K$ is the number of clusters, and $d$ is the feature dimension, a nonlinear mapping $\phi(\cdot)$ projects $\mathbf{X}$ into a reproducing kernel Hilbert space. The kernel matrix is:

$$\mathbf{K}_{i,j} = \kappa(\boldsymbol{x}_i, \boldsymbol{x}_j) = \phi^\top(\boldsymbol{x}_i)\phi(\boldsymbol{x}_j), \quad (1)$$

where $\kappa(\cdot, \cdot)$ is a positive semi-definite kernel function, such as Gaussian (Babaud et al., 1986) or polynomial (Smola

et al., 1998) kernels. The quality of the kernel matrix significantly influences clustering performance, making kernel selection a key challenge. MKC addresses this by combining multiple kernels to generate $N_k$ kernel matrices $\{\mathbf{K}_k\}_{k=1}^{N_k}$.

Multi-Kernel $K$-Means (MKKM) (Huang et al., 2012), a powerful MKC method, assigns weights $\boldsymbol{\gamma} \in \mathbb{R}^{N_k}$ to each kernel and optimizes the following objective:

$$\begin{aligned} \min_{\mathbf{H}, \boldsymbol{\gamma}} \ &\mathrm{Tr}\left(\mathbf{K}_{\boldsymbol{\gamma}}(\mathbf{I}_N - \mathbf{H}\mathbf{H}^\top)\right), \\ \text{s.t. } &\mathbf{H}^\top \mathbf{H} = \mathbf{I}_K, \ \boldsymbol{\gamma}^\top \mathbf{1} = 1, \ \boldsymbol{\gamma} \ge 0, \end{aligned} \quad (2)$$

where $\mathbf{I}_K$ is a $K \times K$ identity matrix, $\mathbf{K}_{\boldsymbol{\gamma}} = \sum_{k=1}^{N_k} \gamma_k \mathbf{K}_k$ represents the weighted kernel combination, and $\mathbf{H} \in \mathbb{R}^{N \times K}$ is obtained through eigen decomposition of $\mathbf{K}_{\boldsymbol{\gamma}}$.

MKC is particularly well-suited for multi-view data, where a single object is represented by features from diverse sources. For multi-view data $\mathcal{X} = \{\mathbf{X}_1, \ldots, \mathbf{X}_{N_d}\}$, applying multiple kernel functions to each view yields $N_k = N_d \times |\mathcal{K}|$ kernel matrices, where $N_d$ is the number of views and $|\mathcal{K}|$ is the number of kernel functions.

### 2.2. Adversarial Attacks on Clustering

Research on adversarial attacks against clustering has predominantly focused on traditional methods operating in the original feature space. Early studies demonstrated that adversarially positioning samples near cluster boundaries can lead to mis-clustering (Dutrisac & Skillicorn, 2008; Skillicorn, 2009). For example, Biggio *et al.* (2013) attacked hierarchical clustering, while Crussell *et al.* (2015) disrupted DBSCAN by merging clusters. Similarly, Chhabra *et al.* (2020) highlighted $K$-Means' vulnerability to boundary perturbations, and Cinà (2022) proposed a black-box attack using genetic algorithms to generate perturbations without knowledge of the clustering algorithm.

Adversarial attacks on MKC, however, remain underexplored. Huang *et al.* (2024) introduced adversarial methods for multi-view clustering using generative adversarial networks (GANs) to disrupt view complementarity and consistency. However, their approach is inapplicable to MKC due to the absence of feedback mechanisms required to optimize GAN parameters. This gap underscores the need for novel methods to systematically evaluate the adversarial robustness of MKC methods.

Due to space constraints, related work on RL for adversarial attacks is discussed in Appendix 3.

## 3. Reinforcement Learning on Adversarial Attacks

Reinforcement learning (RL) has shown success in tackling challenges across various domains, including adversarial ro-

bustness. Its strength lies in optimizing sequential decision-making processes. In adversarial contexts, RL has been applied to attack diverse models and datasets. For instance, Sun *et al.* (2020) injected nodes and adjusted links in graphs to test graph neural networks, while Yang *et al.* (2020) used RL to generate textured patches disrupting convolutional neural networks. Sarkar *et al.* (2023) identified sensitive image regions for minimal perturbations, and Ju *et al.* (2023) leveraged RL to inject malicious nodes into graphs. Despite these successes, RL has not been applied to adversarial attacks on MKC, presenting an open research opportunity to explore assessing MKC robustness.

## 4. Threat Model

Adversarial attacks require clearly defining the attacker's and defender's knowledge, capabilities, and objectives.

**Attacker's Knowledge and Abilities**. This study adopts a realistic *black-box* scenario, where the attacker has no access to the internal workings of the victim MKC method $\mathcal{C}$, including its objective functions and optimization strategies (Chhabra et al., 2020; Cinà et al., 2022). The attacker can access a subset of the original dataset $\mathcal{X}$, introduce perturbations, and observe the resulting representations.

**Defender's Knowledge and Abilities**. The defender lacks specific knowledge of the adversarial attack, such as the identities of perturbed samples or modified kernel matrices. However, they can employ robust MKC methods to mitigate the impact of adversarial perturbations.

**Attacker's Goals**. The attacker aims to introduce minimal perturbations $\mathcal{N}$ to the dataset, constrained by a predefined noise threshold $\epsilon$. These perturbations are designed to significantly degrade clustering performance, as evaluated by external metrics $\mathcal{M}$. The optimization problem is:

$$\underset{\mathcal{N}}{\arg\min}\ \mathcal{M}\left(\mathcal{C}(\mathcal{X} \oplus \mathcal{N}), \mathbf{Y}\right), \quad \text{s.t.} \quad \|\mathcal{N}\| \leq \epsilon, \quad (3)$$

where $\mathbf{Y}$ denotes the ground-truth labels, and $\oplus$ represents the operator that injects adversarial perturbations into specific views of selected samples.

## 5. Methodology

### 5.1. Overview

Adversarial attacks on MKC pose unique challenges due to the characteristics of MKC algorithms: 1) MKC often relies on the $K$-Means algorithm to cluster latent representations without supervision. The absence of labeled data impedes the consistent learning of representative features, while random initialization of cluster centers further exacerbates instability (Lu et al., 2024), complicating the evaluation of perturbation effectiveness. 2) MKC algorithms

require iterative representation learning following each perturbation, significantly increasing computational demands for attackers optimizing perturbations based on previous results. To address these challenges, AdvMKC leverages the PPO algorithm (Schulman et al., 2017; Lee et al., 2024) with an advantage function to mitigate clustering instability. It employs a generator-clusterer RL framework, where the generator produces adversarial perturbations, and the clusterer approximates the victim MKC algorithm, ensuring reliable feedback while reducing computational complexity.

As shown in Figure 1, AdvMKC operates as follows: Given the original dataset $\mathcal{X}$, the attacker perturbs $N_p$ data samples in specific views, acknowledging the practical constraints of multi-view data collection from diverse sources. Section 6 analyzes the impact of varying the number of perturbed samples and views on attack performance. In the RL framework, an episode represents a complete interaction sequence between the adversarial generator and its environment, starting from an initial state and terminating at a predefined state. During each episode, the generator incrementally injects perturbations into the dataset, evaluates rewards based on clustering performance, and adjusts perturbation directions. The process is limited to $T$ steps, defining the episode length of the adversarial attack.

### 5.2. Attack Environment

The adversarial attack is modeled as an RL process, denoted as $\langle \hat{\mathcal{X}}, \mathfrak{N}, \mathfrak{R} \rangle$, with its components detailed below:

**Perturbed Dataset** $\mathfrak{X}$. The perturbed dataset $\mathcal{X}_t$ at time step $t$ represents the adversarially modified multi-view data.

**Perturbation** $\mathfrak{N}$. The attacker is constrained to perturbing $N_p$ samples. To maximize attack efficacy, the MKC method is first applied to the original dataset to derive latent representations: $\mathbf{H} = \mathcal{C}(\mathcal{X})$, where $\mathcal{C}(\cdot)$ represents the MKC algorithm, and $\mathbf{H} \in \mathbb{R}^{N \times K}$ denotes the representations. Subsequently, $K$-Means clustering is performed on $\mathbf{H}$, producing: centroid matrix $\mathbf{M} \in \mathbb{R}^{K \times K}$, with rows $\boldsymbol{m}_i$ as cluster centroids, and cluster indicator matrix $\mathbf{I} \in \{0,1\}^{N \times K}$, where assignments are determined as:

$$\mathbf{I}_{i,k} = \begin{cases} 1, & \text{if } k = \underset{j \in \{1, \cdots, K\}}{\arg\min} \ \|\boldsymbol{h}_i - \boldsymbol{m}_j\|^2; \\ 0, & \text{otherwise}, \end{cases} \quad (4)$$

The $K$-Means algorithm minimizes the clustering error:

$$\underset{\mathbf{M}, \mathbf{I}}{\arg\min} \ \|\mathbf{H} - \mathbf{IM}\|_F^2, \quad s.t. \ \sum_{k=1}^{K} \mathbf{I}_{i,k} = 1. \quad (5)$$

To identify $N^p$ samples for perturbation, the distances between samples and their cluster centroids are computed as:

$$\boldsymbol{d} = \sqrt{\text{diag}\left((\mathbf{H} - \mathbf{IM})(\mathbf{H} - \mathbf{IM})^\top\right)}, \quad (6)$$

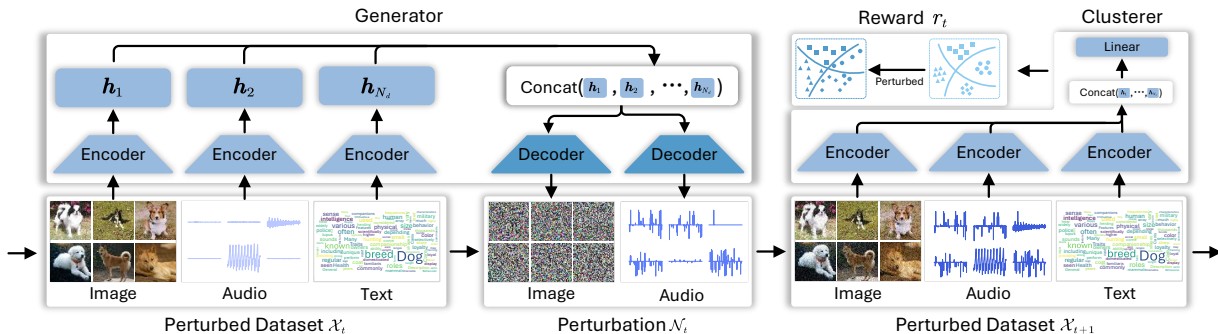

Figure 1. Overview of AdvMKC. Assuming a dataset with three views, i.e., image, audio, and text, the generator is trained to produce adversarial perturbations for the image and audio views. The clusterer evaluates the clustering performance of the perturbed dataset and provides a reward signal to guide the generator's optimization process.

where $\text{diag}(\cdot)$ extracts the diagonal elements of the resulting matrix. The samples with the largest distances are selected:

$$\mathcal{I} = \arg\max_{\mathcal{I} \subset \{1, \cdots, N\}, |\mathcal{I}| = N^p} \sum_{i \in \mathcal{I}} d_i. \tag{7}$$

Given practical constraints, the attacker perturbs only $N_d^p$ views of the selected $N^p$ samples. At time step $t$, adversarial perturbations $\mathcal{N}_t = \{\mathbf{N}_1, \ldots, \mathbf{N}_{N_d^p}\}$ are injected as:

$$\mathbf{X}_{t,i} = \begin{cases} \mathbf{X}_i, & \text{if } i \notin \{1, \ldots, N_d^p\}; \\ \mathbf{X}_i + \mathbf{N}_i, & \text{otherwise.} \end{cases} \tag{8}$$

Here, $\mathcal{N}_t = \text{Decom}(\boldsymbol{n}_t; \mathcal{I}, N_d^p)$, where $\text{Decom}(\cdot)$ distributes vectorized perturbations across views and samples.

**Reward $\mathcal{R}$.** The reward function quantifies the effectiveness of adversarial perturbations and facilitates RL convergence. Since MKC operates in an unsupervised setting without ground-truth labels, the optimization in Eq. (3) is impractical, and gradient-free metrics, such as distance measures (Biggio et al., 2013), are inadequate for guiding perturbation generation. To address these limitations, clustering performance is assessed using the average cosine similarity of data samples within the same cluster prior to the attack. At time step $t$, the reward is defined as:

$$r_t = \frac{1}{N^2} \sum_{i,j} \frac{\mathbf{H}_{i,:}^{(t)} \mathbf{H}_{j,:}^{(t)^\top}}{\|\mathbf{H}_{i,:}^{(t)}\|_2 \|\mathbf{H}_{j,:}^{(t)}\|_2} \odot \mathbf{C}_{i,j}, \tag{9}$$

where $\mathbf{H}_{i,:}^{(t)}$ represents the $i$-th row of the learned representations $\mathbf{H}$ at time step $t$, $\mathbf{C} = \mathbf{II}^\top \in \{0,1\}^{N \times N}$ is the co-occurrence matrix indicating sample cluster assignments, and $\odot$ denotes the Hadamard product.

### 5.3. Generator-Clusterer Framework

To reduce queries to the victim MKC and enhance optimization efficiency, we propose a generator-clusterer framework for perturbation generation and approximation of the victim MKC's functionality.

**Generator.** The generator $\mathcal{G}$ produces adversarial perturbations based on the current perturbed dataset $\mathcal{X}_t$. To ensure the perturbations remain within a predefined threshold $\epsilon$, their magnitude is constrained as:

$$\boldsymbol{n}_t = \epsilon \cdot \mathcal{G}(\mathcal{X}_t; \theta) \cdot \|\mathcal{G}(\mathcal{X}_t; \theta)\|^{-1}, \tag{10}$$

where $\theta$ represents the generator's parameters. The perturbation vector $\boldsymbol{n}_t$ is decomposed into per-view components, as shown in Eq. (8).

The generator $\mathcal{G}$ adopts an encoder-decoder architecture, as shown in Figure 1. The encoder consists of $N_d$ encoders, each corresponding to a dataset view, minimizing computational complexity. The decoder includes $N_d^p$ decoders, one for each perturbed view. Representations from the encoders are concatenated via the $\text{Concat}(\cdot)$ operator, effectively integrating multi-view information for generating perturbations.

**Clusterer.** Following the generation of perturbations by $\mathcal{G}$ and the creation of the updated dataset $\mathcal{X}_t$, the clusterer $\hat{\mathcal{C}}$ approximates clustering performance to provide feedback for optimizing the generator.

The clusterer is designed to mimic the functionality of the victim MKC method while treating it as a black box. Sharing the same encoder architecture as $\mathcal{G}$, the clusterer extracts latent multi-view representations, which are further processed through a multi-layer perceptron to produce clustering outcomes. The clusterer predicts a reward $S(\mathcal{X}_t; \rho)$ for the generated adversarial perturbations using Eq. (9), where $\rho$ represents the clusterer's parameters, and $\mathbf{H}^{(t)}$ is derived from the clusterer instead of the black-box MKC.

### 5.4. Training Procedure

To optimize the adversarial generator and clusterer, we adopt the PPO algorithm (Schulman et al., 2017) with an experience replay mechanism utilizing a memory buffer $\mathcal{B}$. During training, decision-making processes are simulated to generate training data, which is stored in $\mathcal{B}$ with a maximum sequence length of $T$. Each entry in $\mathcal{B}$ is represented as

a triplet $(\mathcal{X}_t, \mathcal{N}_t, r_t)$, encapsulating the perturbed dataset, perturbation, and reward. To address the variability in rewards, the advantage function $A_t$ is employed to estimate the relative value of rewards compared to a baseline:

$$A_t = \sum_{t'=t}^{T} \gamma^{t'-t} r_{t'} - S(\mathcal{X}_t; \rho), \quad (11)$$

where $\gamma$ is the discount factor, and $T$ is the total number of time steps per episode.

**Generator Loss**. The generator aims to maximize the expected advantage while ensuring stable updates through a clipped surrogate objective:

$$\mathcal{L}_g(\theta) = \mathbb{E}_t \left[ \min \left( \vartheta_t A_t, \text{clip}(\vartheta_t, 1 - \eta, 1 + \eta) A_t \right) \right], \quad (12)$$

where $\vartheta_t = \frac{\pi(\mathcal{G}(\mathcal{X}_t; \theta))}{\pi(\mathcal{G}(\mathcal{X}_t; \theta_{\text{old}}))}$ is the probability ratio of the new generator relative to the old one, and $\eta$ is a hyperparameter regulating the allowable deviation. The clipping function $\text{clip}(\cdot)$ constrains the ratio to $(1-\eta, 1+\eta)$, mitigating abrupt updates to enhance the robustness and sample efficiency compared to standard gradient methods.

**Clusterer Loss**. The clusterer estimates the expected attack performance for each perturbation by minimizing the error between predicted rewards and actual outcomes:

$$\mathcal{L}_c(\rho) = \mathbb{E}_t \left[ \left\| S(\mathcal{X}_t; \rho) - \sum_{t'=t}^{T} \gamma^{t'-t} r_{t'} \right\|^2 \right], \quad (13)$$

where $S(\mathcal{X}_t; \rho)$ denotes the clusterer's predicted reward for the perturbed dataset $\mathcal{X}_t$ given parameters $\rho$.

**Entropy Loss**. To foster exploration and prevent premature convergence, an entropy term is added to the loss:

$$\mathcal{L}_{\text{entropy}}(\theta) = -\sum_{\mathcal{X}} \pi \left( \mathcal{G}(\mathcal{X}_t; \theta) \right) \log \pi \left( \mathcal{G}(\mathcal{X}_t; \theta) \right). \quad (14)$$

This term encourages diverse perturbation generation, reducing the risk of suboptimal solutions.

**Magnitude Loss**. To prevent the magnitude of generated perturbations from exceeding a specified limit, a regularization term is introduced to limit the perturbation magnitude:

$$\mathcal{L}_{\text{mag}}(\theta) = \max(\|\mathcal{G}(\mathcal{X}_t; \theta)\| - \epsilon, 0). \quad (15)$$

By optimizing these components together, the training process achieves a balance between exploitation and exploration, refining the generator-clusterer framework. The overall loss function is:

$$\mathcal{L} = -\mathcal{L}_g(\theta) + \alpha \mathcal{L}_c(\rho) - \beta \mathcal{L}_{\text{entropy}}(\theta) + \gamma \mathcal{L}_{\text{mag}}(\theta), \quad (16)$$

where $\alpha$, $\beta$, and $\gamma$ are balancing coefficients.

## 6. Theoretical Analysis

This section explores the impact of injected perturbations on clustering performance, with a focus on the MKKM (Huang et al., 2012) algorithm as a representative clustering method. For optimization stability, the kernel matrix is normalized as $\bar{\mathbf{K}}_{\boldsymbol{\gamma}} = \frac{1}{N} \mathbf{K}_{\boldsymbol{\gamma}}$.

We reformulate the MKKM objective function and describe its optimization using the reduced gradient method. The objective function is defined as:

$$f(\boldsymbol{\gamma}) = \min_{\mathbf{H}^\top \mathbf{H} = \mathbf{I}_K} \text{Tr} \left( \bar{\mathbf{K}}_{\boldsymbol{\gamma}} (\mathbf{I}_N - \mathbf{H}\mathbf{H}^\top) \right). \quad (17)$$

**Proposition 6.1.** *The function $f(\boldsymbol{\gamma})$ in Eq. (17) is differentiable. The gradient's $p$-th component is given by:*

$$\frac{\partial f(\boldsymbol{\gamma})}{\partial \gamma_p} = 2\gamma_p \text{Tr}(\bar{\mathbf{K}}_p (\mathbf{I}_N - \widehat{\mathbf{H}}\widehat{\mathbf{H}}^\top)),$$

*where $\widehat{\mathbf{H}} = \arg\min_{\mathbf{H}^\top \mathbf{H} = \mathbf{I}_K} \text{Tr}(\bar{\mathbf{K}}_{\boldsymbol{\gamma}} (\mathbf{I}_N - \mathbf{H}\mathbf{H}^\top))$.*

Building on Proposition 6.1, we use the reduced gradient method (Rakotomamonjy et al., 2008) to optimize Eq. (17), maintaining the constraints on $\boldsymbol{\gamma}$. Let $u \in \{1, \cdots, N_k\}$ denote a fixed index. The reduced gradient is calculated as:

$$[\nabla f]_p = \frac{\partial f(\boldsymbol{\gamma})}{\partial \gamma_p} - \frac{\partial f(\boldsymbol{\gamma})}{\partial \gamma_u}, \forall\, p \neq u;$$
$$[\nabla f]_u = \sum_{p \neq u} \left( \frac{\partial f(\boldsymbol{\gamma})}{\partial \gamma_u} - \frac{\partial f(\boldsymbol{\gamma})}{\partial \gamma_p} \right). \quad (18)$$

To ensure the positivity of $\boldsymbol{\gamma}$, the descent direction $\boldsymbol{d} = [d_1, \cdots, d_{N_k}]^\top$ is defined as:

$$d_p = \begin{cases} 0, & \text{if } \gamma_p = 0 \text{ and } \kappa(\boldsymbol{\gamma}) > 0; \\ -\frac{1}{N_k - 1} \kappa(\boldsymbol{\gamma}), & \text{if } \gamma_p > 0 \text{ and } p \neq u; \\ \frac{1}{N_k - 1} \sum_{p \neq u, \gamma_p > 0} \kappa(\boldsymbol{\gamma}), & \text{if } p = u; \end{cases} \quad (19)$$

where $\kappa(\boldsymbol{\gamma}) = \frac{\partial f(\boldsymbol{\gamma})}{\partial \gamma_p} - \frac{\partial f(\boldsymbol{\gamma})}{\partial \gamma_u}$.

Compared to Rakotomamonjy *et al.* (2008), the reduced gradient here is normalized by $N_k - 1$ to promote convergence within fewer iterations. The update rule is $\boldsymbol{\gamma} = \boldsymbol{\gamma} + \eta \boldsymbol{d}$, where the step size $\eta$ ensures monotonic convergence of $f(\boldsymbol{\gamma})$ by the reduced gradient method. This algorithm is referred to as MKKM-RG.

**Assumption 6.2.** For any kernel weights $\boldsymbol{\gamma} \in \mathbb{R}^{N_k}$, let $\delta_k(\boldsymbol{\gamma})$ denote the gap between the $k$-th and $(k+1)$-th eigenvalues of $\bar{\mathbf{K}}_{\boldsymbol{\gamma}}$. We assume for any $\boldsymbol{\gamma} \in \mathbf{R}^{N_k}$, $\delta_k(\boldsymbol{\gamma}) \geq c > 0$, where $c$ is a constant.

This assumption is consistent with standard results in matrix theory (Stewart, 1990; Yu et al., 2014) and statistical theory (Von Luxburg et al., 2008; Mitz & Shkolnisky, 2022; Liang et al., 2024a).

*Table 1.* Effectiveness of AdvMKC against traditional MKC methods (%). Lower metric values indicate better performance. *No-attack* denotes the scenario where no adversarial attack is applied during clustering.

| Dataset | Attack | SMKKM | | | | EEOMVC | | | | LSWMKC | | | | LSMKC | | | |
|---|---|---|---|---|---|---|---|---|---|---|---|---|---|---|---|---|---|
| | | NMI | ARI | ACC | PR | NMI | ARI | ACC | PR | NMI | ARI | ACC | PR | NMI | ARI | ACC | PR |
| MSRCv1 | *no-attack* | 59.96 | 50.54 | 69.05 | 69.05 | 65.76 | 59.21 | 76.19 | 76.19 | 59.52 | 47.99 | 67.62 | 69.05 | 59.52 | 47.99 | 67.62 | 69.05 |
| | RAMKC | 61.89 | 53.89 | 72.38 | 72.38 | 58.92 | 47.73 | 63.81 | 66.67 | 55.36 | 43.06 | 64.90 | 65.33 | 58.13 | 49.28 | 69.05 | 69.05 |
| | EAMKC | 60.45 | 52.43 | 71.43 | 71.43 | 56.65 | 46.89 | 64.29 | 66.19 | 55.10 | 43.08 | 64.76 | 64.76 | 54.41 | 43.39 | 64.76 | 64.76 |
| | AdvMKC | **59.44** | **49.57** | **68.00** | **68.00** | **53.18** | **43.03** | **62.86** | **64.29** | **54.22** | **42.42** | **64.29** | **64.29** | **53.90** | **42.05** | **63.19** | **63.19** |
| BBCSport | *no-attack* | 67.50 | 66.21 | 85.11 | 85.11 | **62.50** | 59.39 | 79.78 | 79.78 | 83.12 | 84.42 | 93.75 | 93.75 | 66.30 | 64.96 | 81.07 | 81.07 |
| | RAMKC | 67.50 | 66.21 | 85.11 | 85.11 | 63.21 | 59.70 | 79.96 | 79.96 | 83.12 | 84.42 | 93.75 | 93.75 | 66.27 | 64.92 | 80.88 | 80.88 |
| | EAMKC | 67.50 | 66.21 | 85.11 | 85.11 | 62.99 | 59.29 | 79.78 | 79.78 | 82.47 | 84.08 | 93.57 | 93.57 | 66.27 | 64.92 | 80.88 | 80.88 |
| | AdvMKC | **67.23** | **65.80** | **84.93** | **84.93** | **62.99** | **59.29** | **79.78** | **79.78** | **82.06** | **83.57** | **93.13** | **93.13** | **66.03** | **64.20** | **79.23** | **79.23** |
| Protein Fold | *no-attack* | 39.97 | 11.20 | 29.11 | 38.90 | 42.49 | 12.69 | 32.71 | 40.78 | 39.97 | 11.20 | 29.11 | 38.90 | 40.58 | 11.71 | 30.84 | 38.18 |
| | RAMKC | 31.47 | 09.07 | 23.92 | 29.11 | 42.41 | 12.94 | 33.57 | 39.34 | 42.60 | 13.94 | 31.70 | 38.90 | 39.95 | 11.27 | 30.84 | 37.03 |
| | EAMKC | 29.42 | 07.57 | 21.18 | 26.80 | 41.98 | 13.10 | 32.85 | 40.35 | 43.57 | 15.32 | 33.14 | 39.63 | 39.97 | 11.81 | 31.56 | 38.62 |
| | AdvMKC | **28.68** | **07.06** | **21.09** | **26.22** | **39.15** | **09.66** | **28.67** | **38.33** | **38.93** | **10.81** | **28.27** | **37.61** | **39.90** | **10.99** | **28.10** | **36.75** |
| Caltech 101-7 | *no-attack* | 34.73 | 22.41 | 36.97 | 80.66 | 45.82 | 27.26 | 36.69 | 83.65 | 41.73 | 25.65 | 37.04 | 82.02 | 45.78 | 34.68 | 47.76 | 81.61 |
| | RAMKC | 34.73 | 22.46 | 37.31 | 80.66 | 43.88 | 29.83 | 46.00 | 79.82 | 41.37 | 25.65 | 37.04 | 82.02 | 45.77 | 34.54 | 47.69 | 81.61 |
| | EAMKC | 34.73 | 22.46 | 37.31 | 80.66 | 45.62 | 31.46 | 45.66 | 81.48 | 41.37 | 25.65 | 37.04 | 82.02 | 45.36 | 34.53 | 47.96 | 81.61 |
| | AdvMKC | **34.52** | **22.29** | **36.52** | **80.46** | **43.73** | **26.77** | **32.40** | **77.95** | **41.23** | **25.34** | **37.02** | **81.98** | **45.28** | **34.42** | **47.56** | **81.52** |
| Citeseer | *no-attack* | 25.96 | 23.65 | 51.78 | 54.50 | 23.74 | 18.06 | 44.75 | 46.74 | 41.28 | 41.67 | 66.33 | 69.05 | 19.33 | 14.45 | 41.12 | 41.88 |
| | RAMKC | 27.78 | 25.05 | 52.57 | 56.40 | 22.23 | 17.13 | 41.85 | 44.87 | 41.18 | 41.52 | 66.27 | 68.96 | 16.07 | 12.54 | 37.14 | 38.59 |
| | EAMKC | 26.75 | 23.99 | 51.81 | 55.10 | 22.30 | 16.94 | 41.58 | 45.14 | 40.81 | 41.14 | 66.00 | 68.75 | 23.32 | 18.73 | 43.81 | 46.35 |
| | AdvMKC | **25.33** | **23.54** | **51.30** | **54.38** | **22.02** | **17.03** | **40.63** | **43.08** | **40.73** | **41.07** | **65.06** | **67.81** | **14.27** | **10.39** | **33.81** | **34.29** |
| NUS-WIDE-SCENE | *no-attack* | 08.51 | 04.26 | 24.76 | 32.82 | 07.66 | 03.38 | 21.39 | 30.53 | 06.64 | 02.90 | 23.83 | 29.65 | 08.09 | 04.22 | 22.91 | 31.53 |
| | RAMKC | 08.46 | 04.21 | 24.66 | 32.82 | 07.41 | 02.99 | 20.78 | 30.40 | 06.54 | 02.92 | 23.66 | 29.65 | 08.07 | 04.14 | 23.13 | 31.60 |
| | EAMKC | 08.47 | 04.23 | 24.69 | 32.77 | 06.83 | 03.63 | 23.71 | 32.01 | 06.55 | 02.86 | 23.69 | 29.67 | 08.01 | 04.18 | 22.78 | 31.50 |
| | AdvMKC | **08.39** | **04.14** | **24.51** | **32.40** | **06.68** | **02.62** | **19.44** | **30.31** | **06.50** | **02.58** | **23.55** | **29.51** | **08.00** | **04.00** | **22.53** | **31.35** |

**Theorem 6.3.** *Let $\gamma$ and $\widetilde{\gamma}$ denote the kernel weights obtained from MKKM-RG using the original and perturbed kernel matrices, respectively. Assuming identical initialization and a step size $\eta \leq c$, where $c > 0$, we have:*

$$\|\gamma - \widetilde{\gamma}\|_\infty \precsim \max_{k \in \{1,\cdots,N_k\}} \|\mathbf{N}_k\|,$$

*where $\mathbf{N}_k$ represents the adversarial noise injected into the $k$-th kernel matrix, and $\|\cdot\|$ is the spectral norm.*

From Theorem 6.3, it follows that the impact of adversarial perturbations is determined by both the magnitude and number of perturbations. Specifically, increasing either the perturbation magnitude or the proportion of noisy samples leads to a decline in clustering performance, as further validated by the experimental results in Subsection 7.5.

## 7. Experimental Evaluation

This section presents a comprehensive evaluation to address the following research questions:

- **RQ I**: Can AdvMKC effectively generate perturbations that degrade the performance of MKC methods?

- **RQ II**: Can AdvMKC successfully deceive robust MKC methods designed to handle noisy data?

- **RQ III**: Do perturbations generated by AdvMKC for one MKC method transfer effectively to other MKC methods?

- **RQ IV**: Does AdvMKC demonstrate consistent performance across different settings and hyperparameters?

### 7.1. Experimental Setup

Comprehensive descriptions of the following datasets, baselines, and MKC algorithms are provided in Appendix B. The source code is publicly available at https://github.com/csyuhao/AdvMKC-Official.

**Datasets**. We assess the attack performance of AdvMKC on seven benchmark datasets: MSRCv1 (Winn & Jojic, 2005), BBCSport (Greene & Cunningham, 2006), Protein-Fold (Damoulas & Girolami, 2008), HW-6Views (Huang et al., 2020), Caltech101-7 (Dueck & Frey, 2007), Citeseer (Giles et al., 1998), and NUS-WIDE-SCENE (Chua et al., 2009).

**Compared Methods**. As there are no existing adversarial attack approaches for MKC, we introduce two baseline attack strategies: 1) *RAMKC* adds random Gaussian noise as adversarial perturbations. 2) *EAMKC* optimizes the noise distribution using the Evaluation Strategy (Loshchilov, 2017) and the reward function in Eq. (9).

**MKC Methods**. We compare AdvMKC with seven tra-

*Table 2.* Robustness of AdvMKC against robust MKC methods (%). Lower metric values indicate better performance. *No-attack* refers to clustering without adversarial attacks.

| Dataset | Attack | JMKSC | | | | ONMSC | | | | MKCDNM | | | | MKSSC-ERC | | | |
|---|---|---|---|---|---|---|---|---|---|---|---|---|---|---|---|---|---|
| | | NMI | ARI | ACC | PR | NMI | ARI | ACC | PR | NMI | ARI | ACC | PR | NMI | ARI | ACC | PR |
| MSRCv1 | *no-attack* | 04.87 | 00.11 | 23.81 | 24.29 | 81.64 | 78.16 | 90.00 | 90.00 | 89.77 | 88.41 | 94.76 | 94.76 | 58.24 | 40.12 | 57.14 | 63.81 |
| | RAMKC | 03.42 | -0.45 | 21.90 | 22.86 | 80.62 | 76.99 | 89.52 | 89.52 | 91.74 | 91.13 | 96.19 | 96.19 | 57.06 | 39.18 | 56.19 | 62.38 |
| | EAMKC | 02.84 | -1.06 | 20.00 | 21.90 | 80.19 | 76.19 | 89.05 | 89.05 | 86.17 | 84.93 | 93.33 | 93.33 | 57.06 | 39.18 | 56.19 | 62.38 |
| | AdvMKC | **01.63** | **-1.39** | **19.38** | **19.86** | **80.02** | **76.09** | **89.00** | **89.00** | **85.70** | **82.17** | **90.71** | **90.71** | **56.71** | **38.73** | **55.71** | **61.90** |
| BBCSport | *no-attack* | 43.74 | 23.88 | 55.70 | 56.80 | 80.49 | 81.69 | 88.05 | 88.05 | 46.31 | 39.43 | 59.01 | 66.91 | 39.98 | 32.09 | 54.78 | 58.46 |
| | RAMKC | 53.96 | 37.51 | 65.07 | 65.07 | 80.14 | 81.17 | 87.87 | 87.87 | 46.21 | 39.17 | 58.82 | 66.73 | 39.70 | 25.64 | 47.06 | 59.74 |
| | EAMKC | 46.78 | 25.85 | 55.70 | 59.01 | 80.86 | 81.68 | 88.05 | 88.05 | 45.85 | 38.82 | 58.27 | 66.73 | 39.78 | 25.40 | 46.14 | 59.74 |
| | AdvMKC | **38.74** | **20.30** | **52.21** | **55.51** | **80.04** | **81.02** | **87.35** | **87.35** | **45.11** | **37.86** | **57.90** | **65.99** | **38.77** | **24.90** | **45.06** | **58.19** |
| Protein Fold | *no-attack* | 27.41 | 08.02 | 23.63 | 25.36 | 20.08 | 07.09 | 19.31 | 20.61 | 18.89 | 04.46 | 20.32 | 21.90 | 22.06 | 07.38 | 22.33 | 23.34 |
| | RAMKC | 25.25 | 07.91 | 21.76 | 23.78 | 21.23 | 06.49 | 18.44 | 21.04 | 18.54 | 05.22 | 21.33 | 21.76 | 22.49 | 07.33 | 20.89 | 21.76 |
| | EAMKC | 24.80 | 07.81 | 22.19 | 23.63 | 20.65 | 07.40 | 20.03 | 21.04 | 19.29 | 04.19 | 21.76 | 21.76 | 14.48 | 01.32 | 17.29 | **17.44** |
| | AdvMKC | **23.65** | **06.90** | **21.04** | **22.92** | **19.48** | **06.46** | **18.16** | **20.46** | **18.08** | **04.14** | **20.03** | **21.05** | **14.04** | **01.19** | **17.15** | 17.58 |
| Caltech 101-7 | *no-attack* | 00.98 | -0.23 | 38.60 | 54.21 | 42.65 | 26.65 | 37.99 | 82.43 | 50.76 | 32.93 | 48.85 | 85.28 | 55.31 | 40.14 | 51.76 | 84.40 |
| | RAMKC | 01.12 | 00.08 | 17.84 | 51.14 | 42.66 | 26.69 | 38.20 | 82.43 | 50.32 | 32.89 | 45.93 | 85.01 | 55.44 | 40.41 | 52.10 | 84.46 |
| | EAMKC | 00.94 | 00.86 | 29.51 | 54.14 | 42.66 | 26.66 | 38.06 | 82.43 | 50.49 | 32.78 | 49.19 | 85.21 | 55.88 | 40.18 | 51.76 | **83.92** |
| | AdvMKC | **00.30** | **-0.30** | **17.31** | **50.14** | **42.52** | **26.60** | **36.90** | **82.40** | **50.25** | **32.22** | **44.71** | 84.87 | **55.15** | **38.78** | **49.59** | 83.92 |
| Citeseer | *no-attack* | 01.15 | -0.04 | 21.50 | 21.68 | 24.20 | 21.07 | 50.09 | 51.78 | 21.06 | 19.57 | 46.35 | 48.70 | 38.54 | 34.42 | 61.62 | 63.22 |
| | RAMKC | 01.58 | 00.96 | 22.95 | 24.15 | 26.39 | 24.76 | 54.35 | 56.28 | 19.70 | 16.11 | 43.51 | 48.10 | 39.50 | 36.57 | 61.14 | 65.34 |
| | EAMKC | 01.59 | 01.00 | 23.07 | 24.18 | 27.04 | 25.24 | 54.71 | 56.73 | 07.55 | 02.49 | 27.57 | 31.31 | 39.96 | 36.81 | 62.32 | 65.31 |
| | AdvMKC | **01.06** | **-0.05** | **21.07** | **21.21** | **24.00** | **20.38** | **47.19** | **49.94** | **06.91** | **02.47** | **27.14** | **30.40** | **38.01** | **33.55** | **60.91** | **62.46** |
| NUS-WIDE-SCENE | *no-attack* | 01.11 | 00.23 | 21.27 | 28.47 | 07.87 | 03.10 | 21.03 | 29.99 | 07.93 | 03.39 | 21.27 | 30.55 | 06.32 | 02.97 | 21.78 | 30.84 |
| | RAMKC | 01.11 | 00.40 | **17.34** | 28.47 | 07.88 | 03.10 | 21.15 | 29.99 | 07.93 | 03.40 | 21.39 | 30.55 | 06.38 | 03.00 | 21.73 | 30.92 |
| | EAMKC | 01.36 | -0.44 | 18.58 | 28.47 | 07.95 | 03.10 | 21.10 | 29.96 | 07.93 | 03.41 | 21.20 | 30.62 | 06.38 | 02.97 | 21.71 | 30.87 |
| | AdvMKC | **00.40** | **-0.52** | 18.01 | **27.68** | **07.81** | **03.06** | **20.15** | **29.01** | **07.87** | **03.30** | **21.12** | **30.31** | **06.30** | **02.95** | **21.70** | **30.82** |

ditional MKC methods: MVC-LFA (Wang et al., 2019), LSMKKM (Liu et al., 2021), MKKM-SR (Lu et al., 2022), SMKKM (Liu, 2023), EEOMVC (Wang et al., 2024), LSWMKC (Li et al., 2024), and LSMKC (Liang et al., 2024b), as well as four robust MKC methods designed for noisy data: JMKSC (Yang et al., 2019), ONMSC (Zhou et al., 2020), MKCDNM (Zhang et al., 2022), and MKSSC-ERC (Xu et al., 2024).

**Evaluation Metrics**. To assess clustering performance before and after adversarial attacks, we use four external evaluation metrics: normalized mutual information (NMI), adjusted rand index (ARI), accuracy (ACC), and purity (PR).

**Implementation Details**. To ensure imperceptible adversarial perturbations, we modify 10% of the dataset samples, with 50% of the views being targeted by default. Following established practices (Chen et al., 2020), the perturbation magnitude is computed using the $\ell_2$ norm, with a default value of $\epsilon = 0.1\sqrt{d}$. Specifically, for a feature vector of size $d$, an $\ell_2$ norm perturbation of magnitude $\epsilon$ corresponds to an average perturbation of $\epsilon/\sqrt{d}$ per feature.

### 7.2. RQ I: Effectiveness Evaluation

This section evaluates the adversarial robustness of state-of-the-art MKC methods, targeting seven methods as victims. Results for SMKKM, EEOMVC, LSWMKC, and LSMKC

are presented in Table 1, with additional findings provided in Table 5. Key observations are summarized as follows:

1) **Adversarial attacks degrade MKC performance.** As demonstrated in the theoretical analysis (Section 6) and experimental results (Table 1), adversarial perturbations significantly reduce clustering performance. For instance, on the MSRCv1 dataset, attacking EEOMVC reduces NMI from 65.76% to 53.18%. These results confirm that MKC methods, like other machine learning models, are susceptible to adversarial attacks.

2) **ACC and PR metrics coincide in some cases.** Since clustering is an unsupervised task, both metrics rely on the best possible label assignment. ACC determines the optimal one-to-one mapping between cluster IDs and ground truth labels using the Hungarian algorithm (Kuhn, 2010), whereas PR assigns each cluster to the most frequent ground truth label. When clusters closely align with single ground truth classes (i.e., each cluster predominantly contains data points from one class), both metrics yield identical values.

3) **AdvMKC outperforms RAMKC and EAMKC.** AdvMKC consistently achieves superior performance in generating effective perturbations. For example, on the MSRCv1 dataset, when attacking SMKKM, AdvMKC achieves the most significant impact, demonstrating its ability to optimize the perturbation generator through RL.

## 7.3. RQ II: Robustness Evaluation

This section evaluates the effectiveness of AdvMKC against robust MKC algorithms, specifically targeting JMKSC, ON-MSC, MKCDNM, and MKSSC-ERC as victim methods. The experimental results, summarized in Table 2, yield the following key insights:

1) **Robust MKC methods struggle to defend against adversarial attacks.** For instance, on the MSRCv1 dataset, when attacking the JMKSC algorithm, AdvMKC reduces NMI to 1.63%. This highlights a critical limitation of current robust MKC methods, which often assume perturbations follow Gaussian distributions, rendering them ineffective against more complex adversarial scenarios.

2) **Robust MKC methods are resilient to Gaussian noise.** For example, on the ProteinFold dataset, RAMKC does not significantly degrade the clustering performance of the ONMSC method (NMI: 21.23% under attack vs. 20.08% without attack). This demonstrates that robust MKC methods are well-suited to handle Gaussian noise but remain vulnerable to advanced adversarial attacks.

## 7.4. RQ III: Transferability Evaluation

This subsection investigates the transferability of adversarial attacks, acknowledging that the defender's specific MKC method is often unknown. It also considers an extreme scenario in which attacking a multi-view clustering model necessitates extensive queries to the target MKC method, leading to substantial resource consumption. To evaluate this, a randomly selected MKC method (referred to as the *surrogate MKC*) is used to train the perturbation generator, and its effectiveness is subsequently tested on a different victim MKC method. The experimental results, summarized in Table 3, yield the following key observation:

1) **Adversarial attacks demonstrate significant transferability across MKC methods.** For instance, when JMKSC is used as the surrogate MKC method and SMKKM is the victim method, AdvMKC still achieves a 5% reduction in NMI. This transferability likely arises from the common characteristics underlying clustering performance in different unsupervised learning approaches.

## 7.5. RQ IV: Sensitivity Evaluation

This subsection investigates the influence of three factors on attack performance: the number of selected samples $N_d^p$, the number of perturbed views $N_k^p$, and the magnitude of adversarial perturbations $\epsilon$. LSMKC and ONMSC are chosen as the victim MKC methods. Note that *in scenarios where PR and ACC are nearly identical, the ACC curve is obscured by the PR curve in the experimental results*.

**Number of Selected Samples** $N_d^p$. The impact of the

*Table 3.* Transferability of AdvMKC by leveraging surrogate MKC methods to attack unknown victim MKC methods.

| Dataset | Surrogate MKC | SMKKM | | | | MKSSC-ERC | | | |
|---|---|---|---|---|---|---|---|---|---|
| | | NMI | ARI | ACC | PR | NMI | ARI | ACC | PR |
| MSRCv1 | *no-attack* | 59.96 | 50.54 | 69.05 | 69.05 | 58.24 | 40.12 | 57.14 | 63.81 |
| | EEOMVC | 57.65 | 49.61 | 68.48 | 68.48 | 56.14 | 38.22 | 57.02 | 62.86 |
| | LSMKC | 57.35 | 48.73 | 68.52 | 68.52 | 55.43 | 39.20 | 57.10 | 61.43 |
| | JMKSC | 54.39 | 45.58 | 64.29 | 64.29 | 54.96 | 37.81 | 55.71 | 59.52 |
| | ONMSC | 56.37 | 48.67 | 68.05 | 68.05 | 54.19 | 37.19 | 56.14 | 60.48 |
| BBCSport | *no-attack* | 67.50 | 66.21 | 85.11 | 85.11 | 39.98 | 32.09 | 54.78 | 58.46 |
| | EEOMVC | 66.25 | 65.03 | 84.56 | 84.56 | 38.85 | 30.75 | 52.12 | 57.68 |
| | LSMKC | 66.57 | 65.50 | 84.74 | 84.74 | 38.22 | 26.42 | 47.46 | 57.29 |
| | JMKSC | 66.89 | 65.35 | 84.74 | 84.74 | 32.67 | 09.37 | 71.73 | 45.40 |
| | ONMSC | 67.42 | 66.12 | 85.00 | 85.00 | 39.52 | 24.94 | 46.32 | 58.82 |
| Protein Fold | *no-attack* | 39.97 | 11.20 | 29.11 | 38.90 | 22.06 | 07.38 | 22.33 | 23.34 |
| | EEOMVC | 30.15 | 07.54 | 21.33 | 26.80 | 21.38 | 01.37 | 18.59 | 22.91 |
| | LSMKC | 29.57 | 08.11 | 20.89 | 26.37 | 22.01 | 00.75 | 17.29 | 20.75 |
| | JMKSC | 28.93 | 07.40 | 20.61 | 24.64 | 21.78 | 02.77 | 19.88 | 22.97 |
| | ONMSC | 29.88 | 07.75 | 21.33 | 27.81 | 21.45 | 02.98 | 18.30 | 22.36 |

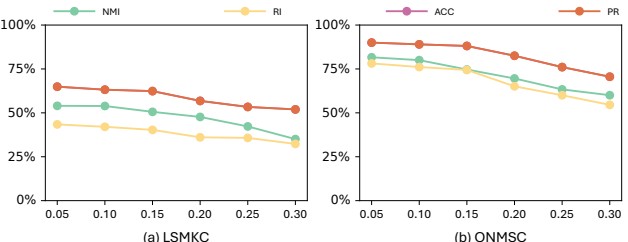

*Figure 2.* The impact of the proportion of perturbed samples on the MSRCv1 dataset.

proportion of perturbed samples is analyzed by varying the ratio of selected samples to the total dataset as $\{0.05, 0.10, 0.15, 0.20, 0.25, 0.30\}$. Evaluations are conducted on the MSRCv1, BBCSport, and ProteinFold datasets. Results for the MSRCv1 dataset are shown in Figure 2, while Figures 5 and 6 present results for BBC-Sport and ProteinFold, respectively. The findings indicate that increasing the proportion of perturbed samples leads to a corresponding decline in clustering performance, as expected, since perturbing more samples exacerbates performance degradation.

**Number of Perturbed Views** $N_k^p$. The effect of the number of perturbed views is evaluated on the MSRCv1, HW-6Views, and Caltech101-7 datasets, with $N_k^p$ varied as $\{1, 2, 3, 4, 5, 6\}$. Results for the MSRCv1 dataset are provided in Figure 3, while Figures 7 and 8 show results for HW-6Views and Caltech101-7, respectively. The results demonstrate that an increase in the number of perturbed views leads to a notable decline in clustering performance.

**Magnitude of Adversarial Perturbations** $\epsilon$. The influence of perturbation magnitude is assessed by varying $\epsilon$ as $\{0.01\sqrt{d}, 0.05\sqrt{d}, 0.10\sqrt{d}, 0.15\sqrt{d}, 0.20\sqrt{d}, 0.25\sqrt{d}\}$ on the MSRCv1, BBCSport, and ProteinFold datasets. Results for the MSRCv1 dataset are shown in Figure 4, with Figures 9 and 10 displaying results for the other two datasets,

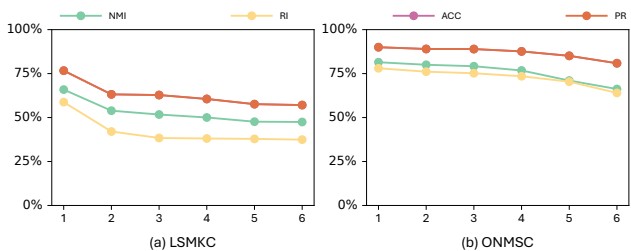

*Figure 3.* The impact of the number of perturbed views on the MSRCv1 dataset.

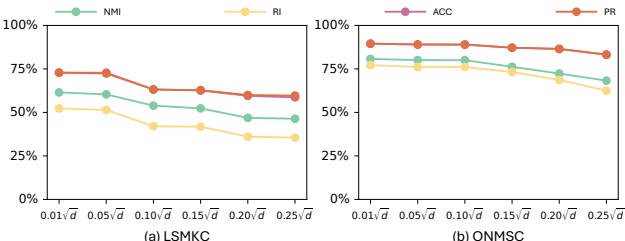

*Figure 4.* The impact of adversarial perturbation magnitude on the MSRCv1 dataset.

respectively. The analysis reveals that higher magnitudes cause more severe degradation in clustering performance.

## 8. Conclusion

In this paper, we proposed AdvMKC, the first black-box adversarial attack method specifically designed for MKC. By formulating adversarial attacks on MKC as an RL problem, we utilized proximal policy optimization with an advantage function to address the instability of clustering results during optimization. We developed an innovative generator-clusterer framework, where the generator produces adversarial perturbations and the clusterer approximates MKC behavior, thereby significantly reducing computational overhead. A theoretical analysis was conducted to clarify the impact of injected perturbations on clustering performance, supported by extensive evaluations. Experimental results validated the effectiveness, robustness, and transferability of the proposed approach.

## Acknowledgement

This work is supported by the National Science Fund for Distinguished Young Scholars of China (No. 62325604), and the National Natural Science Foundation of China (No. 62441618 and 62276271).

## Impact Statement

This paper presents work whose goal is to advance the field of Machine Learning. There are many potential societal consequences of our work, none of which we feel must be specifically highlighted here.

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

# A. Theoritical Analysis

## A.1. Proof of Proposition 6.1

**Proposition A.1.** *The function $f(\boldsymbol{\gamma})$ in Eq. (17) is differentiable. The $p$-th component of its gradient is given by:*

$$\frac{\partial f(\boldsymbol{\gamma})}{\partial \gamma_p} = 2\gamma_p \operatorname{Tr}(\bar{\mathbf{K}}_p(\mathbf{I}_N - \widehat{\mathbf{H}}\widehat{\mathbf{H}}^\top)),$$

*where $\widehat{\mathbf{H}} = \arg\min_{\mathbf{H}^\top\mathbf{H}=\mathbf{I}_K} \operatorname{Tr}(\bar{\mathbf{K}}_{\boldsymbol{\gamma}}(\mathbf{I}_N - \mathbf{H}\mathbf{H}^\top)).$*

*Proof.* The objective function $f(\boldsymbol{\gamma})$ can be expressed as the sum of two components:

$$f(\boldsymbol{\gamma}) = f_1(\boldsymbol{\gamma}) + f_2(\boldsymbol{\gamma}),$$

where

$$f_1(\boldsymbol{\gamma}) = \operatorname{Tr}(\bar{\mathbf{K}}_{\boldsymbol{\gamma}}), \quad f_2(\boldsymbol{\gamma}) = \max_{\mathbf{H}^\top\mathbf{H}=\mathbf{I}_K} \operatorname{Tr}(\bar{\mathbf{K}}_{\boldsymbol{\gamma}}\mathbf{H}\mathbf{H}^\top).$$

**1) Differentiability of $f_1(\boldsymbol{\gamma})$.** It is straightforward to verify that $f_1(\boldsymbol{\gamma})$ is differentiable, and its gradient is given by:

$$\frac{\partial f_1(\boldsymbol{\gamma})}{\partial \gamma_p} = 2\gamma_p \operatorname{Tr}(\bar{\mathbf{K}}_p).$$

**2) Differentiability of $f_2(\boldsymbol{\gamma})$.** From Theorem 1 in Liu *et al.* (2023), $f_2(\boldsymbol{\gamma})$ is differentiable, and its gradient is:

$$\frac{\partial f_2(\boldsymbol{\gamma})}{\partial \gamma_p} = 2\gamma_p \operatorname{Tr}(\bar{\mathbf{K}}_p\widehat{\mathbf{H}}\widehat{\mathbf{H}}^\top),$$

where $\widehat{\mathbf{H}} = \arg\max_{\mathbf{H}^\top\mathbf{H}=\mathbf{I}_K} \operatorname{Tr}(\bar{\mathbf{K}}_{\boldsymbol{\gamma}}\mathbf{H}\mathbf{H}^\top).$

**3) Differentiability of $f(\boldsymbol{\gamma})$.** Since $f(\boldsymbol{\gamma}) = f_1(\boldsymbol{\gamma}) + f_2(\boldsymbol{\gamma})$, it is differentiable, and its gradient is the sum of the gradients of $f_1(\boldsymbol{\gamma})$ and $f_2(\boldsymbol{\gamma})$:

$$\frac{\partial f(\boldsymbol{\gamma})}{\partial \gamma_p} = \frac{\partial f_1(\boldsymbol{\gamma})}{\partial \gamma_p} + \frac{\partial f_2(\boldsymbol{\gamma})}{\partial \gamma_p}.$$

By substituting the expressions for $\frac{\partial f_1(\boldsymbol{\gamma})}{\partial \gamma_p}$ and $\frac{\partial f_2(\boldsymbol{\gamma})}{\partial \gamma_p}$, we obtain:

$$\frac{\partial f(\boldsymbol{\gamma})}{\partial \gamma_p} = 2\gamma_p \operatorname{Tr}\left(\bar{\mathbf{K}}_p(\mathbf{I}_N - \widehat{\mathbf{H}}\widehat{\mathbf{H}}^\top)\right).$$

Furthermore, $\widehat{\mathbf{H}}$ is also the optimal solution of the equivalent minimization problem: $\widehat{\mathbf{H}} = \arg\min_{\mathbf{H}^\top\mathbf{H}=\mathbf{I}_K} \operatorname{Tr}(\bar{\mathbf{K}}_{\boldsymbol{\gamma}}(\mathbf{I}_N - \mathbf{H}\mathbf{H}^\top)).$ $\qquad\square$

## A.2. Proof of Thereom 6.3

Before completing the proof of Theorem 6.3, we introduce the following definition, which quantifies the alignment level between a base kernel matrix and the consensus clustering indicator matrix.

**Definition A.2.** Let $\boldsymbol{\alpha} \in \mathbb{R}^{N_k}$ represent a set of kernel weights, and let $\widehat{\mathbf{H}}_{\boldsymbol{\alpha}}$ denote the consensus clustering indicator matrix. The alignment level between the $p$-th base kernel $\bar{\mathbf{K}}_p$ and $\widehat{\mathbf{H}}_{\boldsymbol{\alpha}}$ is defined as:

$$\mathcal{T}(\bar{\mathbf{K}}_p, \widehat{\mathbf{H}}_{\boldsymbol{\alpha}}) = \operatorname{Tr}(\bar{\mathbf{K}}_p) - \operatorname{Tr}(\bar{\mathbf{K}}_p\widehat{\mathbf{H}}_{\boldsymbol{\alpha}}\widehat{\mathbf{H}}_{\boldsymbol{\alpha}}^\top).$$

To prove Theorem 6.3, we utilize the following three lemmas. Here, we provide the proof of Lemma A.3.

**Lemma A.3.** *Let $\mathbf{A}, \mathbf{B} \in \mathbb{R}^{N \times N}$ be positive semi-definite matrices. Then,*

$$\operatorname{Tr}(\mathbf{A}\mathbf{B}) \le \|\mathbf{A}\| \cdot \operatorname{Tr}(\mathbf{B}),$$

*where $\|\mathbf{A}\|$ denotes the spectral norm of $\mathbf{A}$.*

*Proof.* Let $\mathbf{B}^{\frac{1}{2}}$ be the unique positive semi-definite matrix satisfying $\mathbf{B}^{\frac{1}{2}}\mathbf{B}^{\frac{1}{2}} = \mathbf{B}$. Using this, we rewrite the trace term as:

$$\mathrm{Tr}(\mathbf{A}\mathbf{B}) = \mathrm{Tr}\left(\mathbf{B}^{\frac{1}{2}}\mathbf{A}\mathbf{B}^{\frac{1}{2}}\right).$$

Since $\mathbf{B}^{\frac{1}{2}}(\|\mathbf{A}\| \cdot \mathbf{I}_N)\mathbf{B}^{\frac{1}{2}} \succcurlyeq \mathbf{B}^{\frac{1}{2}}\mathbf{A}\mathbf{B}^{\frac{1}{2}}$ (by the definition of the spectral norm), we have:

$$\mathrm{Tr}\left(\mathbf{B}^{\frac{1}{2}}\mathbf{A}\mathbf{B}^{\frac{1}{2}}\right) \leq \mathrm{Tr}\left(\mathbf{B}^{\frac{1}{2}}(\|\mathbf{A}\| \cdot \mathbf{I}_N)\mathbf{B}^{\frac{1}{2}}\right).$$

Expanding the right-hand side gives:

$$\mathrm{Tr}\left(\mathbf{B}^{\frac{1}{2}}(\|\mathbf{A}\| \cdot \mathbf{I}_N)\mathbf{B}^{\frac{1}{2}}\right) = \|\mathbf{A}\| \cdot \mathrm{Tr}(\mathbf{B}),$$

which concludes the proof. $\qquad\square$

The following lemma provides a perturbation bound for eigenvectors of Hermitian matrices, which is useful in our analysis.

**Lemma A.4.** *(Yu et al., 2014) Let $\mathbf{A}, \mathbf{B} \in \mathbb{R}^{N \times N}$ be Hermitian matrices with eigenvalues $\lambda_1 \geq \cdots \geq \lambda_N$ and $\widehat{\lambda}_1 \geq \cdots \geq \widehat{\lambda}_N$, respectively. Fix $1 \leq r \leq s \leq N$, and assume $\min(\lambda_{r-1} - \lambda_r, \lambda_s - \lambda_{s+1}) > 0$, where $\lambda_0 := \infty$ and $\lambda_{N+1} := -\infty$. Define $d := s - r + 1$, and let $\mathbf{H} = [\boldsymbol{h}_r, \boldsymbol{h}_{r+1}, \ldots, \boldsymbol{h}_s] \in \mathbb{R}^{N \times d}$ and $\widehat{\mathbf{H}} = [\widehat{\boldsymbol{h}}_r, \widehat{\boldsymbol{h}}_{r+1}, \ldots, \widehat{\boldsymbol{h}}_s] \in \mathbb{R}^{N \times d}$ be column-orthogonal matrices satisfying $\mathbf{A}\boldsymbol{h}_j = \lambda_j \boldsymbol{h}_j$ and $\mathbf{B}\widehat{\boldsymbol{h}}_j = \widehat{\lambda}_j \widehat{\boldsymbol{h}}_j$ for $j \in \{r, \ldots, s\}$. Then, the following bound holds:*

$$\left\|\sin\theta(\mathbf{H}, \widehat{\mathbf{H}})\right\|_{\mathrm{F}} \leq \frac{2\min(d^{1/2}\|\mathbf{A} - \mathbf{B}\|_{\mathrm{op}}, \|\mathbf{A} - \mathbf{B}\|_{\mathrm{F}})}{\min(\lambda_{r-1} - \lambda_r, \lambda_s - \lambda_{s+1})},$$

*where $\theta(\mathbf{H}, \widehat{\mathbf{H}}) \in \mathbb{R}^{d \times d}$ is diagonal with $j$-th diagonal entry equal to $\arccos(\boldsymbol{h}_j^\top \widehat{\boldsymbol{h}}_j)$. Additionally, there exists an orthogonal matrix $\widehat{\mathbf{O}} \in \mathbb{R}^{d \times d}$ such that:*

$$\left\|\widehat{\mathbf{O}}\widehat{\mathbf{H}} - \mathbf{H}\right\|_{\mathrm{F}} \leq \frac{2^{3/2}\min(d^{1/2}\|\mathbf{A} - \mathbf{B}\|_{\mathrm{op}}, \|\mathbf{A} - \mathbf{B}\|_{\mathrm{F}})}{\min(\lambda_{r-1} - \lambda_r, \lambda_s - \lambda_{s+1})},$$

*where $\|\cdot\|_{\mathrm{F}}$ represents the Frobenius norm and $\|\cdot\|_{\mathrm{op}}$ denotes the operator norm.*

Without loss of generality, assume the attacker injects adversarial perturbations $\mathcal{N} = \{\mathbf{N}_1, \ldots, \mathbf{N}_{N_k^p}\}$ into the first $N_k^p$ views, while the remaining views $\{N_k^p + 1, \ldots, N_k\}$ remain unperturbed. The perturbed kernel matrices are given by $\widetilde{\mathcal{K}} = \{\widetilde{\mathbf{K}}_1, \ldots, \widetilde{\mathbf{K}}_{N_k^p}, \mathbf{K}_{N_k^p+1}, \ldots, \mathbf{K}_{N_k}\}$, and the normalized fused kernel matrix is:

$$\overline{\mathbf{K}} = \frac{1}{N}\widetilde{\mathbf{K}}.$$

**Lemma A.5.** *For any two sets of kernel weights $\boldsymbol{\alpha}$ and $\boldsymbol{\beta}$, let:*

$$\widehat{\mathbf{H}}_{\boldsymbol{\alpha}} = \underset{\mathbf{H}^\top\mathbf{H}=\mathbf{I}_K}{\arg\min} \mathrm{Tr}(\overline{\mathbf{K}}_{\boldsymbol{\alpha}}(\mathbf{I}_N - \mathbf{H}\mathbf{H}^\top)), \quad \widetilde{\mathbf{H}}_{\boldsymbol{\beta}} = \underset{\mathbf{H}^\top\mathbf{H}=\mathbf{I}_K}{\arg\min} \mathrm{Tr}(\overline{\mathbf{K}}_{\boldsymbol{\beta}}(\mathbf{I}_N - \mathbf{H}\mathbf{H}^\top)).$$

*Then, for $p \in \{1, \ldots, N_k^p\}$, the difference between $\mathcal{T}(\overline{\mathbf{K}}_p, \widehat{\mathbf{H}}_{\boldsymbol{\alpha}})$ and $\mathcal{T}(\overline{\mathbf{K}}_p, \widetilde{\mathbf{H}}_{\boldsymbol{\beta}})$ satisfies:*

$$\left|\mathcal{T}(\overline{\mathbf{K}}_p, \widehat{\mathbf{H}}_{\boldsymbol{\alpha}}) - \mathcal{T}(\overline{\mathbf{K}}_p, \widetilde{\mathbf{H}}_{\boldsymbol{\beta}})\right| \precsim \|\boldsymbol{\alpha} - \boldsymbol{\beta}\|_\infty + \max_{q \in \{1, \ldots, N_k\}} \|\mathbf{N}_q\|.$$

*Similarly, for $p \in \{N_k^p + 1, \ldots, N_k\}$, the bound is given by:*

$$\mathcal{T}(\overline{\mathbf{K}}_p, \widehat{\mathbf{H}}_{\boldsymbol{\alpha}}) - \mathcal{T}(\overline{\mathbf{K}}_p, \widetilde{\mathbf{H}}_{\boldsymbol{\beta}}) \precsim \|\boldsymbol{\alpha} - \boldsymbol{\beta}\|_\infty + \max_{q \in \{1, \ldots, N_k\}} \|\mathbf{N}_q\|.$$

*Proof.* For $p \in \{1, \cdots, N_k\}$, consider the following:

$$
\begin{aligned}
&\left| \mathcal{T}(\bar{\mathbf{K}}_p, \widehat{\mathbf{H}}_{\boldsymbol{\alpha}}) - \mathcal{T}(\overline{\mathbf{K}}_p, \widetilde{\mathbf{H}}_{\boldsymbol{\beta}}) \right| \\
&= \left| \mathrm{Tr}\left( \bar{\mathbf{K}}_p (\mathbf{I}_N - \widehat{\mathbf{H}}_{\boldsymbol{\alpha}} \widehat{\mathbf{H}}_{\boldsymbol{\alpha}}^\top) \right) - \mathrm{Tr}\left( \overline{\mathbf{K}}_p (\mathbf{I}_N - \widetilde{\mathbf{H}}_{\boldsymbol{\beta}} \widetilde{\mathbf{H}}_{\boldsymbol{\beta}}^\top) \right) \right| \\
&\leq \|\mathbf{N}_p\| \cdot \mathrm{Tr}\left( \frac{1}{N} \mathbf{I}_N \right) + \left| \mathrm{Tr}(\bar{\mathbf{K}}_p \widehat{\mathbf{H}}_{\boldsymbol{\alpha}} \widehat{\mathbf{H}}_{\boldsymbol{\alpha}}^\top) - \mathrm{Tr}(\bar{\mathbf{K}}_p \widehat{\mathbf{H}}_{\boldsymbol{\beta}} \widehat{\mathbf{H}}_{\boldsymbol{\beta}}^\top) \right| + \left| \mathrm{Tr}(\bar{\mathbf{K}}_p \widehat{\mathbf{H}}_{\boldsymbol{\beta}} \widehat{\mathbf{H}}_{\boldsymbol{\beta}}^\top) - \mathrm{Tr}(\bar{\mathbf{K}}_p \widetilde{\mathbf{H}}_{\boldsymbol{\beta}} \widetilde{\mathbf{H}}_{\boldsymbol{\beta}}^\top) \right| \\
&\quad + \left| \mathrm{Tr}(\bar{\mathbf{K}}_p \widetilde{\mathbf{H}}_{\boldsymbol{\beta}} \widetilde{\mathbf{H}}_{\boldsymbol{\beta}}^\top) - \mathrm{Tr}(\overline{\mathbf{K}}_p \widetilde{\mathbf{H}}_{\boldsymbol{\beta}} \widetilde{\mathbf{H}}_{\boldsymbol{\beta}}^\top) \right|
\end{aligned}
\tag{20}
$$

Applying the trace inequality and Lemma A.3, we bound each term:

$$
\begin{aligned}
&\left| \mathcal{T}(\bar{\mathbf{K}}_p, \widehat{\mathbf{H}}_{\boldsymbol{\alpha}}) - \mathcal{T}(\overline{\mathbf{K}}_p, \widetilde{\mathbf{H}}_{\boldsymbol{\beta}}) \right| \\
&\leq \|\mathbf{N}_p\| + \|\bar{\mathbf{K}}_p\|_{\mathrm{F}} \cdot \left\| \widehat{\mathbf{H}}_{\boldsymbol{\alpha}} \widehat{\mathbf{H}}_{\boldsymbol{\alpha}}^\top - \widehat{\mathbf{H}}_{\boldsymbol{\beta}} \widehat{\mathbf{H}}_{\boldsymbol{\beta}}^\top \right\|_{\mathrm{F}} + \|\bar{\mathbf{K}}_p\|_{\mathrm{F}} \cdot \left\| \widehat{\mathbf{H}}_{\boldsymbol{\beta}} \widehat{\mathbf{H}}_{\boldsymbol{\beta}}^\top - \widetilde{\mathbf{H}}_{\boldsymbol{\beta}} \widetilde{\mathbf{H}}_{\boldsymbol{\beta}}^\top \right\|_{\mathrm{F}} + \|\bar{\mathbf{K}}_p - \overline{\mathbf{K}}_p\| \cdot \left| \frac{1}{N} \mathrm{Tr}(\widetilde{\mathbf{H}}_{\boldsymbol{\beta}} \widetilde{\mathbf{H}}_{\boldsymbol{\beta}}^\top) \right|
\end{aligned}
\tag{21}
$$

Using Jensen's inequality, we have $\|\bar{\mathbf{K}}_p\|_{\mathrm{F}} \leq 1$, i.e.,

$$
\|\bar{\mathbf{K}}_p\|_{\mathrm{F}} = \sqrt{\frac{1}{N^2} \mathrm{Tr}(\mathbf{K}_p^2)} \leq \sqrt{\frac{1}{N^2} \mathrm{Tr}^2(\mathbf{K}_p)} \leq 1.
\tag{22}
$$

Subsequently, applying the above equation, we have

$$
\left| \mathcal{T}(\bar{\mathbf{K}}_p, \widehat{\mathbf{H}}_{\boldsymbol{\alpha}}) - \mathcal{T}(\overline{\mathbf{K}}_p, \widetilde{\mathbf{H}}_{\boldsymbol{\beta}}) \right| \precsim \|\mathbf{N}_p\| + \left\| \widehat{\mathbf{H}}_{\boldsymbol{\alpha}} \widehat{\mathbf{H}}_{\boldsymbol{\alpha}}^\top - \widehat{\mathbf{H}}_{\boldsymbol{\beta}} \widehat{\mathbf{H}}_{\boldsymbol{\beta}}^\top \right\|_{\mathrm{F}} + \left\| \widehat{\mathbf{H}}_{\boldsymbol{\beta}} \widehat{\mathbf{H}}_{\boldsymbol{\beta}}^\top - \widetilde{\mathbf{H}}_{\boldsymbol{\beta}} \widetilde{\mathbf{H}}_{\boldsymbol{\beta}}^\top \right\|_{\mathrm{F}}.
\tag{23}
$$

For any orthogonal matrix $\mathbf{O} \in \mathbb{R}^{K \times K}$, we have

$$
\begin{aligned}
\left\| \widehat{\mathbf{H}}_{\boldsymbol{\alpha}} \widehat{\mathbf{H}}_{\boldsymbol{\alpha}}^\top - \widehat{\mathbf{H}}_{\boldsymbol{\beta}} \widehat{\mathbf{H}}_{\boldsymbol{\beta}}^\top \right\|_{\mathrm{F}} &= \left\| \widehat{\mathbf{H}}_{\boldsymbol{\alpha}} \mathbf{O} \mathbf{O}^\top \widehat{\mathbf{H}}_{\boldsymbol{\alpha}}^\top - \widehat{\mathbf{H}}_{\boldsymbol{\beta}} \widehat{\mathbf{H}}_{\boldsymbol{\beta}}^\top \right\|_{\mathrm{F}} \\
&\leq \left\| \widehat{\mathbf{H}}_{\boldsymbol{\alpha}} \mathbf{O} \mathbf{O}^\top \widehat{\mathbf{H}}_{\boldsymbol{\alpha}}^\top - \widehat{\mathbf{H}}_{\boldsymbol{\alpha}} \mathbf{O} \widehat{\mathbf{H}}_{\boldsymbol{\beta}}^\top \right\|_{\mathrm{F}} + \left\| \widehat{\mathbf{H}}_{\boldsymbol{\alpha}} \mathbf{O} \widehat{\mathbf{H}}_{\boldsymbol{\beta}}^\top - \widehat{\mathbf{H}}_{\boldsymbol{\beta}} \widehat{\mathbf{H}}_{\boldsymbol{\beta}}^\top \right\|_{\mathrm{F}} \\
&\leq \|\widehat{\mathbf{H}}_{\boldsymbol{\alpha}} \mathbf{O}\| \cdot \left\| \widehat{\mathbf{H}}_{\boldsymbol{\alpha}} \mathbf{O} - \widehat{\mathbf{H}}_{\boldsymbol{\beta}} \right\|_{\mathrm{F}} + \|\widehat{\mathbf{H}}_{\boldsymbol{\beta}}\| \cdot \left\| \widehat{\mathbf{H}}_{\boldsymbol{\alpha}} \mathbf{O} - \widehat{\mathbf{H}}_{\boldsymbol{\beta}} \right\|_{\mathrm{F}} \\
&\precsim \left\| \widehat{\mathbf{H}}_{\boldsymbol{\alpha}} \mathbf{O} - \widehat{\mathbf{H}}_{\boldsymbol{\beta}} \right\|_{\mathrm{F}}.
\end{aligned}
\tag{24}
$$

By Lemma A.4 (let $r = 1, s = k$), with Assumption 6.2, we have

$$
\begin{aligned}
\left\| \widehat{\mathbf{H}}_{\boldsymbol{\alpha}} \widehat{\mathbf{H}}_{\boldsymbol{\alpha}}^\top - \widehat{\mathbf{H}}_{\boldsymbol{\beta}} \widehat{\mathbf{H}}_{\boldsymbol{\beta}}^\top \right\|_{\mathrm{F}} &\precsim \left\| \widetilde{\mathbf{H}}_{\boldsymbol{\alpha}} \mathbf{O} - \widetilde{\mathbf{H}}_{\boldsymbol{\beta}} \right\|_{\mathrm{F}} \\
&\precsim \left\| \frac{1}{N} \mathbf{K}_{\boldsymbol{\alpha}} - \frac{1}{N} \mathbf{K}_{\boldsymbol{\beta}} \right\|_{\mathrm{F}} \\
&\leq \sum_{p=1}^{N_k} |\alpha_p^2 - \beta_p^2| \cdot \left\| \frac{1}{N} \mathbf{K}_p \right\|_{\mathrm{F}} \\
&\leq \|\boldsymbol{\alpha} - \boldsymbol{\beta}\|_\infty \cdot \sum_{p=1}^{N_k} (\alpha_p + \beta_p) \\
&\precsim \|\boldsymbol{\alpha} - \boldsymbol{\beta}\|_\infty.
\end{aligned}
\tag{25}
$$

Similarly, we have

$$
\left\| \widehat{\mathbf{H}}_{\boldsymbol{\beta}} \widehat{\mathbf{H}}_{\boldsymbol{\beta}}^\top - \widetilde{\mathbf{H}}_{\boldsymbol{\beta}} \widetilde{\mathbf{H}}_{\boldsymbol{\beta}}^\top \right\|_{\mathrm{F}} \precsim \left\| \bar{\mathbf{K}}_{\boldsymbol{\beta}} - \overline{\widetilde{\mathbf{K}}}_{\boldsymbol{\beta}} \right\|_{\mathrm{F}} \leq \sum_{p=1}^{N_k^p} \beta_p^2 \left\| \frac{1}{N}(\mathbf{K}_p - \widetilde{\mathbf{K}}_p) \right\|_{\mathrm{F}} \leq \sum_{p=1}^{N_k^p} \beta_p^2 \|\mathbf{N}_p\| \cdot \left\| \frac{1}{N} \mathbf{I} \right\|_{\mathrm{F}} \leq \max_{q \in \{1, \cdots, N_k\}} \|\mathbf{N}_q\|.
\tag{26}
$$

Substituting Eq. (25) and Eq. (26) into Eq. (24), we can obtain the first bound as follows

$$\left|\mathcal{T}(\bar{\mathbf{K}}_p, \widehat{\mathbf{H}}_{\boldsymbol{\alpha}}) - \mathcal{T}(\bar{\mathbf{K}}_p, \widetilde{\mathbf{H}}_{\boldsymbol{\beta}})\right| \precsim \|\boldsymbol{\alpha} - \boldsymbol{\beta}\|_{\infty} + \max_{q \in \{1, \cdots, N_k\}} \|\mathbf{N}_q\|.$$

For $p \in \{N_k^p + 1, \cdots, N_k\}$, we have

$$
\begin{aligned}
\left|\mathcal{T}(\bar{\mathbf{K}}_p, \widehat{\mathbf{H}}_{\boldsymbol{\alpha}}) - \mathcal{T}(\bar{\mathbf{K}}_p, \widetilde{\mathbf{H}}_{\boldsymbol{\beta}})\right| &= \left|\frac{1}{N} \operatorname{Tr}(\mathbf{K}_p(\widetilde{\mathbf{H}}_{\boldsymbol{\beta}}\widetilde{\mathbf{H}}_{\boldsymbol{\beta}}^{\top} - \widehat{\mathbf{H}}_{\boldsymbol{\alpha}}\widehat{\mathbf{H}}_{\boldsymbol{\alpha}}^{\top}))\right| \\
&\leq \left|\frac{1}{N} \operatorname{Tr}(\mathbf{K}_p(\widetilde{\mathbf{H}}_{\boldsymbol{\beta}}\widetilde{\mathbf{H}}_{\boldsymbol{\beta}}^{\top} - \widehat{\mathbf{H}}_{\boldsymbol{\beta}}\widehat{\mathbf{H}}_{\boldsymbol{\beta}}^{\top}))\right| + \left|\frac{1}{N} \operatorname{Tr}(\mathbf{K}_p(\widehat{\mathbf{H}}_{\boldsymbol{\beta}}\widehat{\mathbf{H}}_{\boldsymbol{\beta}}^{\top} - \widehat{\mathbf{H}}_{\boldsymbol{\alpha}}\widehat{\mathbf{H}}_{\boldsymbol{\alpha}}^{\top}))\right| \quad (27) \\
&\precsim \|\boldsymbol{\alpha} - \boldsymbol{\beta}\|_{\infty} + \max_{q \in \{1, \cdots, N_k\}} \|\mathbf{N}_q\|.
\end{aligned}
$$

$\square$

Finally, we present the proof of Theorem 6.3.

*Proof.* Assume noiseless kernel matrices lead to the kernel weight sequence $\{\boldsymbol{\alpha}^{(t)}\}_{t=0}^{T}$, where $\boldsymbol{\alpha}^{(t)}$ represents the weights at the $t$-th update. Similarly, let $\{\boldsymbol{\beta}^{(t)}\}_{t=0}^{T}$ denote the kernel weights for the noised kernel matrices. Since the initialization is identical, we have $\boldsymbol{\alpha}^{(0)} = \boldsymbol{\beta}^{(0)}$.

Fix an index $u \in \{1, \ldots, N_k^p\}$. At the $(t+1)$-th step, we analyze the change in $|\alpha_u^{(t+1)} - \beta_u^{(t+1)}|$:

$$
\begin{aligned}
&|\alpha_u^{(t+1)} - \beta_u^{(t+1)}| - |\alpha_u^{(t)} - \beta_u^{(t)}| \\
\leq\, &|\alpha_u^{(t+1)} - \alpha_u^{(t)} - (\beta_u^{(t+1)} - \beta_u^{(t)})| \\
\leq\, &\frac{1}{N_k - 1} \underbrace{\left| \sum_{p \neq u, p \in \{1, \cdots, N_k^p\}} \left(\alpha_p^{(t)} \mathcal{T}(\mathbf{K}_p, \widehat{\mathbf{H}}_{\boldsymbol{\alpha}^{(t)}}) - \alpha_u^{(t)} \mathcal{T}(\mathbf{K}_u, \widehat{\mathbf{H}}_{\boldsymbol{\alpha}^{(t)}})\right) - \sum_{p \neq u, p \in \{1, \cdots, N_k^p\}} \left(\beta_p^{(t)} \mathcal{T}(\widetilde{\mathbf{K}}_p, \widetilde{\mathbf{H}}_{\boldsymbol{\beta}^{(t)}}) - \beta_u^{(t)} \mathcal{T}(\widetilde{\mathbf{K}}_u, \widetilde{\mathbf{H}}_{\boldsymbol{\beta}^{(t)}})\right) \right|}_{\mathcal{A}} \\
&+ \frac{1}{N_k - 1} \underbrace{\left| \sum_{p = N_k^p + 1}^{m} \left(\alpha_p^{(t)} \mathcal{T}(\mathbf{K}_p, \widehat{\mathbf{H}}_{\boldsymbol{\alpha}^{(t)}}) - \alpha_u^{(t)} \mathcal{T}(\mathbf{K}_u, \widehat{\mathbf{H}}_{\boldsymbol{\alpha}^{(t)}})\right) - \sum_{p = N_k^p + 1}^{N_k} \left(\beta_p^{(t)} \mathcal{T}(\mathbf{K}_p, \widetilde{\mathbf{H}}_{\boldsymbol{\beta}^{(t)}}) - \beta_u^{(t)} \mathcal{T}(\widetilde{\mathbf{K}}_u, \widetilde{\mathbf{H}}_{\boldsymbol{\beta}^{(t)}})\right) \right|}_{\mathcal{B}}.
\end{aligned}
$$

$$(28)$$

For term $\mathcal{A}$, we bound it as follows:

$$
\begin{aligned}
\mathcal{A} =\, &\left| \sum_{p \neq u, p \in \{1, \cdots, N_k^p\}} \left(\alpha_p^{(t)} \mathcal{T}(\mathbf{K}_p, \widehat{\mathbf{H}}_{\boldsymbol{\alpha}^{(t)}}) - \beta_p^{(t)} \mathcal{T}(\widetilde{\mathbf{K}}_p, \widetilde{\mathbf{H}}_{\boldsymbol{\beta}^{(t)}})\right) - (N_k^p - 1)\left(\alpha_u^{(t)} \mathcal{T}(\mathbf{K}_u, \widehat{\mathbf{H}}_{\boldsymbol{\alpha}^{(t)}}) - \beta_u^{(t)} \mathcal{T}(\widetilde{\mathbf{K}}_u, \widetilde{\mathbf{H}}_{\boldsymbol{\beta}^{(t)}})\right) \right| \\
\precsim\, &(N_k^p - 1) \max_{q \in \{1, \cdots, N_k^p\}} \left|\alpha_q^{(t)} \mathcal{T}(\mathbf{K}_q, \widehat{\mathbf{H}}_{\boldsymbol{\alpha}^{(t)}}) - \beta_q^{(t)} \mathcal{T}(\widetilde{\mathbf{K}}_q, \widetilde{\mathbf{H}}_{\boldsymbol{\beta}^{(t)}})\right| \\
=\, &(N_k^p - 1) \max_{q \in \{1, \cdots, N_k^p\}} \left|\alpha_q^{(t)} \mathcal{T}(\mathbf{K}_q, \widehat{\mathbf{H}}_{\boldsymbol{\alpha}^{(t)}}) - \beta_q^{(t)} \mathcal{T}(\mathbf{K}_q, \widehat{\mathbf{H}}_{\boldsymbol{\alpha}^{(t)}}) + \beta_q^{(t)} \mathcal{T}(\mathbf{K}_q, \widehat{\mathbf{H}}_{\boldsymbol{\alpha}^{(t)}}) - \beta_q^{(t)} \mathcal{T}(\widetilde{\mathbf{K}}_q, \widetilde{\mathbf{H}}_{\boldsymbol{\beta}^{(t)}})\right| \\
\leq\, &(N_k^p - 1) \max_{q \in \{1, \cdots, N_k^p\}} \left|\alpha_q^{(t)} - \beta_q^{(t)}\right| \cdot \mathcal{T}(\mathbf{K}_q, \widehat{\mathbf{H}}_{\boldsymbol{\alpha}^{(t)}}) + (N_k^p - 1) \max_{q \in \{1, \cdots, N_k^p\}} \beta_q^{(t)} \cdot \left|\mathcal{T}(\mathbf{K}_q, \widehat{\mathbf{H}}_{\boldsymbol{\alpha}^{(t)}}) - \mathcal{T}(\widetilde{\mathbf{K}}_q, \widetilde{\mathbf{H}}_{\boldsymbol{\beta}^{(t)}})\right| \\
\precsim\, &(N_k^p - 1)\|\boldsymbol{\alpha}^{(t)} - \boldsymbol{\beta}^{(t)}\|_{\infty} + (N_k^p - 1)\left(\|\boldsymbol{\alpha}^{(t)} - \boldsymbol{\beta}^{(t)}\|_{\infty} + \max_{q \in \{1, \cdots, N_k^p\}} \|\mathbf{N}_q\|\right) \\
&\text{(By Lemma A.5.)} \\
\precsim\, &(N_k^p - 1)(\|\boldsymbol{\alpha}^{(t)} - \boldsymbol{\beta}^{(t)}\|_{\infty} + \max_{q \in \{1, \cdots, N_k^p\}} \|\mathbf{N}_q\|)
\end{aligned}
$$

$$(29)$$

where $\|\mathbf{N}_q\|$ denotes the noise magnitude.

For term $\mathcal{B}$, a similar derivation yields:

$$
\mathcal{B} = \left| \sum_{p=N_k^p+1}^{N_k} \left( \alpha_p^{(t)} \mathcal{T}(\mathbf{K}_p, \widehat{\mathbf{H}}_{\boldsymbol{\alpha}^{(t)}}) - \beta_p^{(t)} \mathcal{T}(\mathbf{K}_p, \widetilde{\mathbf{H}}_{\boldsymbol{\beta}^{(t)}}) \right) - (N_k - N_k^p) \left( \alpha_u^{(t)} \mathcal{T}(\mathbf{K}_u, \widehat{\mathbf{H}}_{\boldsymbol{\alpha}^{(t)}}) - \beta_u^{(t)} \mathcal{T}(\widetilde{\mathbf{K}}_u, \widetilde{\mathbf{H}}_{\boldsymbol{\beta}^{(t)}}) \right) \right|
$$

$$
\precsim \max_{q \in \{1, \cdots, N_k\}} (N_k - N_k^p) \left( \alpha_q^{(t)} \mathcal{T}(\mathbf{K}_q, \widehat{\mathbf{H}}_{\boldsymbol{\alpha}^{(t)}}) - \beta_q^{(t)} \mathcal{T}(\mathbf{K}_q, \widetilde{\mathbf{H}}_{\boldsymbol{\beta}^{(t)}}) \right) + (N_k - N_k^p) \left( \alpha_u^{(t)} \mathcal{T}(\mathbf{K}_u, \widehat{\mathbf{H}}_{\boldsymbol{\alpha}^{(t)}}) - \beta_u^{(t)} \mathcal{T}(\widetilde{\mathbf{K}}_u, \widetilde{\mathbf{H}}_{\boldsymbol{\beta}^{(t)}}) \right).
$$

(By Lemma A.5.)

$$
\precsim (N_k - N_k^p) \left( \|\boldsymbol{\alpha}^{(t)} - \boldsymbol{\beta}^{(t)}\|_\infty + \max_{q \in \{1, \cdots, N_k\}} \|\mathbf{N}_q\| \right).
$$

(30)

Combining Eq. (29), Eq. (30), and Eq. (28), we have

$$
|\alpha_u^{(t+1)} - \beta_u^{(t+1)}| - |\alpha_u^{(t)} - \beta_u^{(t)}| \precsim \|\boldsymbol{\alpha}^{(t)} - \boldsymbol{\beta}^{(t)}\|_\infty + \max_{q \in \{1, \cdots, N_k\}} \|\mathbf{N}_q\|.
$$

(31)

Similarly, for $p \in \{1, \cdots, N_k^p\}, p \neq u$, we have

$$
\begin{aligned}
&|\alpha_p^{(t+1)} - \beta_p^{(t+1)}| - |\alpha_p^{(t)} - \beta_p^{(t)}| \\
\leq & |\alpha_p^{(t+1)} - \alpha_p^{(t)} - (\beta_p^{(t+1)} - \beta_p^{(t)})| \\
\leq & \frac{1}{N_k - 1} \left| \alpha_u^{(t)} \mathcal{T}(\mathbf{K}_u, \widehat{\mathbf{H}}_{\boldsymbol{\alpha}^{(t)}}) - \alpha_p^{(t)} \mathcal{T}(\mathbf{K}_p, \widehat{\mathbf{H}}_{\boldsymbol{\alpha}^{(t)}}) \right| + \frac{1}{N_k - 1} \left| \beta_u^{(t)} \mathcal{T}(\widetilde{\mathbf{K}}_u, \widetilde{\mathbf{H}}_{\boldsymbol{\beta}^{(t)}}) - \beta_p^{(t)} \mathcal{T}(\widetilde{\mathbf{K}}_p, \widetilde{\mathbf{H}}_{\boldsymbol{\beta}^{(t)}}) \right| \\
\precsim & \|\boldsymbol{\alpha}^{(t)} - \boldsymbol{\beta}^{(t)}\|_\infty + \max_{q \in \{1, \cdots, N_k\}} \|\mathbf{N}_q\|.
\end{aligned}
$$

(32)

For $p \in \{N_k^p + 1, \cdots, N_k\}$, we have

$$
\begin{aligned}
&|\alpha_p^{(t+1)} - \beta_p^{(t+1)}| - |\alpha_p^{(t)} - \beta_p^{(t)}| \\
\leq & \frac{1}{N_k - 1} \left| \alpha_u^{(t)} \mathcal{T}(\mathbf{K}_u, \widehat{\mathbf{H}}_{\boldsymbol{\alpha}^{(t)}}) - \alpha_p^{(t)} \mathcal{T}(\mathbf{K}_p, \widehat{\mathbf{H}}_{\boldsymbol{\alpha}^{(t)}}) \right| + \frac{1}{N_k - 1} \left| \beta_u^{(t)} \mathcal{T}(\widetilde{\mathbf{K}}_u, \widetilde{\mathbf{H}}_{\boldsymbol{\beta}^{(t)}}) - \beta_p^{(t)} \mathcal{T}(\mathbf{K}_p, \widetilde{\mathbf{H}}_{\boldsymbol{\beta}^{(t)}}) \right| \\
\precsim & \|\boldsymbol{\alpha}^{(t)} - \boldsymbol{\beta}^{(t)}\|_\infty + \max_{q \in \{1, \cdots, N_k\}} \|\mathbf{N}_q\|.
\end{aligned}
$$

(33)

Combining the upper bounds in Eq. (31), Eq. (32) and Eq. (33), we can obtain

$$
\|\boldsymbol{\alpha}^{(t+1)} - \boldsymbol{\beta}^{(t+1)}\|_\infty \precsim \|\boldsymbol{\alpha}^{(t)} - \boldsymbol{\beta}^{(t)}\|_\infty + \max_{q \in \{1, \cdots, N_k\}} \|\mathbf{N}_q\|.
$$

(34)

Based on the above recurrence formula, we have

$$
\begin{aligned}
\|\boldsymbol{\alpha}^{(T)} - \boldsymbol{\beta}^{(T)}\|_\infty &\precsim \|\boldsymbol{\alpha}^{(T-1)} - \boldsymbol{\beta}^{(T-1)}\|_\infty + \max_{q \in \{1, \cdots, N_k\}} \|\mathbf{N}_q\| \\
&\precsim \cdots \precsim \|\boldsymbol{\alpha}^{(0)} - \boldsymbol{\beta}^{(0)}\|_\infty + \max_{q \in \{1, \cdots, N_k\}} \|\mathbf{N}_q\| = \max_{q \in \{1, \cdots, N_k\}} \|\mathbf{N}_q\|.
\end{aligned}
$$

The proof is complete.

$\square$

# B. Details of Experimental Setup

**Datasets**. We evaluate our approach on seven multi-view datasets, described as follows:

- **MSRCv1** (Winn & Jojic, 2005) contains 210 objects from 7 classes (tree, building, airplane, cow, face, car, bicycle) with 6 data views.

Table 4. Summary of seven benchmark datasets.

| Dataset | # Sample | # View | # Cluster | # Feature | Perturbed View Index |
|---|---|---|---|---|---|
| MSRCv1 | 210 | 5 | 7 | [24, 576, 512, 256, 254] | [1, 4] |
| BBCSport | 544 | 2 | 5 | [3183, 3203] | [1] |
| ProteinFold | 694 | 12 | 27 | [27, 27, $\cdots$, 27, 27] | [1, 2, 3, 4] |
| Caltech101-7 | 1474 | 6 | 7 | [48, 40, 254, 1984, 512, 928] | [1, 2] |
| HW-6Views | 2000 | 6 | 10 | [216, 76, 64, 6, 240, 47] | [3, 4] |
| Citeseer | 3312 | 2 | 6 | [3312, 3703] | [1] |
| NUS-WIDE-SCENE | 4095 | 5 | 33 | [128, 73, 144, 225, 64] | [2, 5] |

- **BBCSport** (Greene & Cunningham, 2006) includes 544 sports news articles in 5 categories with two views: 3,183-dimensional MTX features and 3,203-dimensional TERMS features.

- **ProteinFold** (Damoulas & Girolami, 2008) comprises 694 instances across 27 classes, each represented by 12 feature sets.

- **Caltech101-7** (Dueck & Frey, 2007) is a subset of the Caltech101 dataset containing 1,474 images across 7 categories (e.g., DollaBill, Faces). Each image is described using six feature types: Gabor (48 dimensions), Wavelet Transform and Moments (40 dimensions), Centered Histogram (254 dimensions), Histogram of Oriented Gradients (1,984 dimensions), Global Image Scene Representation (512 dimensions), and Local Binary Patterns (928 dimensions).

- **HW-6Views** (Huang et al., 2020) contains 2,000 handwritten numerals (0–9) with 6 heterogeneous views labeled into 10 classes.

- **Citeseer** (Giles et al., 1998) is a graph dataset of 3,312 scientific publications in 6 categories (e.g., AI, ML). Nodes represent papers with a 3,703-dimensional keyword vector, and edges indicate citations.

- **NUS-WIDE-SCENE** (Chua et al., 2009) features 4,095 images across 33 classes with 5 feature types: Color Histogram (64 dimensions), Color Correlations (144 dimensions), Edge Direction Histogram (73 dimensions), Wavelet Texture (128 dimensions), and Block-Wise Color Moment (225 dimensions).

The summary of each dataset is presented in Table 4. In our evaluations, we utilize Gaussian (Babaud et al., 1986) and Polynomial (Smola et al., 1998) kernels to construct the kernel matrices for each data view.

**MKC Methods.** We evaluate our method (AdvMKC) against seven traditional MKC approaches:

- **MVC-LFA** (Wang et al., 2019) aligns consensus clustering partitions with weighted base partitions.

- **LSMKKM** (Liu et al., 2021) incorporates local alignment to enhance the fusion of base kernel information.

- **MKKM-SR** (Lu et al., 2022) combines spectral rotation with multiple kernel $K$-Means clustering for simultaneous optimization of discrete and continuous cluster labels.

- **SMKKM** (Liu, 2023) minimizes kernel weight alignment errors while maximizing clustering assignment accuracy.

- **EEOMVC** (Wang et al., 2024) integrates common latent representation and clustering indicator matrix generation into a unified framework.

- **LSWMKC** (Li et al., 2024) denoises graphs and learns neighborhood kernels to reveal latent local manifold representations.

- **LSMKC** (Liang et al., 2024b) assumes isotropic Gaussian distributions for samples and optimizes cluster assignments of expectation kernel matrices.

Additionally, we compare AdvMKC with four robust MKC methods designed for noisy data:

- **JMKSC** (Yang et al., 2019) learns an optimal consensus kernel from predefined candidates for reliable clustering results.

*Table 5.* Effectiveness of AdvMKC against traditional MKC methods (%). Lower metric values indicate better performance. *No-attack* refers to clustering without adversarial attacks.

| Dataset | Attack | MVC-LFA | | | | LSMKKM | | | | MKKM-SR | | | |
|---|---|---|---|---|---|---|---|---|---|---|---|---|---|
| | | NMI | ARI | ACC | PR | NMI | ARI | ACC | PR | NMI | ARI | ACC | PR |
| MSRCv1 | *no-attack* | 63.42 | 56.26 | 74.76 | 74.76 | 61.38 | 53.92 | 73.81 | 73.81 | 58.54 | 49.99 | 70.00 | 72.38 |
| | RAMKC | 61.75 | 53.58 | 71.43 | 71.90 | 61.75 | 54.51 | 73.81 | 73.81 | 53.95 | 44.29 | 66.19 | 66.19 |
| | EAMKC | 62.29 | 54.05 | 73.33 | 73.33 | 60.16 | 52.48 | 70.48 | 70.48 | 54.54 | 45.19 | 66.67 | 66.67 |
| | AdvMKC | **59.37** | **51.15** | **70.00** | **70.48** | **60.02** | **51.39** | **69.95** | **69.95** | **52.73** | **42.78** | **64.76** | **65.19** |
| BBCSport | *no-attack* | 69.23 | 71.25 | 81.88 | 81.07 | 54.32 | 39.72 | 69.49 | 69.85 | 54.67 | 52.98 | 67.46 | 74.08 |
| | RAMKC | 67.79 | 60.00 | 71.32 | 80.88 | 52.23 | 41.49 | 70.04 | 70.40 | 55.15 | 53.24 | 67.46 | 74.08 |
| | EAMKC | 68.76 | 60.51 | 71.32 | 80.88 | 54.88 | 40.21 | 69.67 | 70.04 | 55.12 | 53.24 | 67.46 | 74.08 |
| | AdvMKC | **68.70** | **60.31** | **71.25** | **80.02** | **29.08** | **13.69** | **51.10** | **51.84** | **52.60** | **51.58** | **66.42** | **72.04** |
| Protein Fold | *no-attack* | 44.07 | 17.47 | 34.01 | 41.21 | 22.76 | 04.72 | 19.60 | 24.64 | 39.97 | 11.20 | 29.11 | 38.90 |
| | RAMKC | 44.15 | 16.78 | 34.29 | 42.51 | 20.73 | 03.35 | 19.16 | 23.05 | 38.64 | 10.33 | 27.95 | 37.32 |
| | EAMKC | 43.75 | 16.56 | 33.86 | 42.22 | 18.79 | 01.80 | 16.71 | 20.46 | 39.88 | 11.28 | 29.39 | 39.34 |
| | AdvMKC | **42.34** | **15.22** | **32.85** | **39.77** | **18.02** | **01.65** | **16.14** | **20.31** | **37.26** | **08.16** | **25.09** | **35.18** |
| Caltech 101-7 | *no-attack* | 48.34 | 28.13 | 37.52 | 83.79 | 55.92 | 39.75 | 52.10 | 85.35 | 43.90 | 33.74 | 48.10 | 79.35 |
| | RAMKC | 45.89 | 27.93 | 38.94 | 82.50 | 56.05 | 39.40 | 52.10 | 85.35 | 44.08 | 33.78 | 48.03 | 79.72 |
| | EAMKC | 45.95 | 27.64 | 36.57 | 82.43 | 55.93 | 39.31 | 52.17 | **85.28** | 42.84 | 29.68 | 38.94 | **44.78** |
| | AdvMKC | **45.78** | **27.63** | **36.43** | **82.29** | **55.91** | **39.26** | **52.04** | **85.28** | **39.30** | **26.22** | **36.25** | 45.69 |
| Citeseer | *no-attack* | 28.70 | 26.30 | 54.50 | 57.07 | 07.06 | 00.08 | 25.45 | 25.51 | 20.73 | 15.64 | 44.20 | 44.96 |
| | RAMKC | 28.40 | 25.98 | 54.26 | 56.88 | 07.00 | 00.07 | 25.39 | 25.45 | 11.20 | 07.26 | 31.91 | 34.21 |
| | EAMKC | 28.36 | 26.03 | 54.41 | 56.94 | 07.18 | 00.11 | 25.54 | 25.60 | **10.17** | 06.93 | 32.37 | 34.72 |
| | AdvMKC | **28.08** | **25.01** | **54.00** | **56.10** | **05.83** | **00.05** | **24.43** | **24.61** | 10.17 | **06.90** | **31.06** | **32.42** |
| NUS-WIDE-SCENE | *no-attack* | 07.67 | 03.75 | 22.98 | **32.36** | 06.34 | 01.28 | 26.01 | 31.16 | 06.56 | 02.75 | 22.69 | 31.79 |
| | RAMKC | 07.90 | 03.90 | 23.81 | 32.63 | 06.23 | 01.22 | 25.59 | 30.96 | 06.59 | 02.88 | 22.93 | 31.89 |
| | EAMKC | 07.53 | 03.68 | 22.95 | 32.45 | 06.32 | 01.16 | 25.64 | 30.99 | 06.57 | 02.81 | 22.78 | 31.84 |
| | AdvMKC | **07.50** | **03.21** | **23.64** | 32.41 | **06.20** | **01.08** | **25.54** | **30.92** | **06.47** | **02.70** | **22.66** | **31.79** |

- **ONMSC** (Zhou et al., 2020) enhances the representational capacity of Laplacian matrices by incorporating high-order connections.

- **MKCDNM** (Zhang et al., 2022) decomposes kernel noise into dual components and minimizes them in a late fusion framework.

- **MKSSC-ERC** (Xu et al., 2024) employs a stable clustering framework with automatic initialization of optimal cluster centers.

**Compared Methods.** As no existing adversarial attack methods are available for MKC, we propose two baseline attack strategies to benchmark AdvMKC:

- *RAMKC* generates Gaussian noise $n$ from $\mathcal{N}(\mathbf{0}, \mathbf{I})$, normalizes it, and rescales its magnitude to satisfy adversarial constraints: $\widetilde{n} = \epsilon \frac{n}{\|n\|}$. The noise $\widetilde{n}$ is injected into $N_d^p$ samples with $N_k^p$ views.

- *EAMKC* formulates adversarial perturbations as a black-box optimization problem (Williams & Li, 2023; Fang et al., 2023; 2022). Using the LM-CMA strategy (Loshchilov, 2017), it optimizes the mean and variance of noise distributions with a reward function defined in Eq. (9).

## C. Additional Experimental Results

**Effectiveness Evaluation**. Table 5 presents the experimental results of adversarial attacks on the MVC-LFA, LSMKKM, and MKKM-SR algorithms. The results reveal the susceptibility of MKC methods to adversarial attacks, with AdvMKC demonstrating superior performance compared to other approaches.

**Sensitivity Evaluation**. We assess the influence of three factors on attack performance: the number of selected samples $N_d^p$,

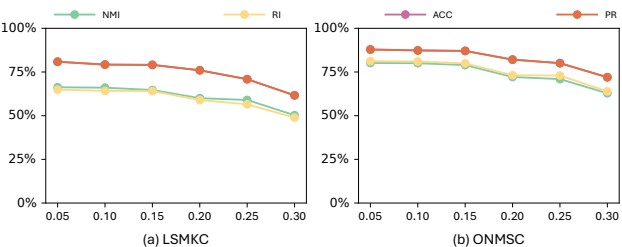

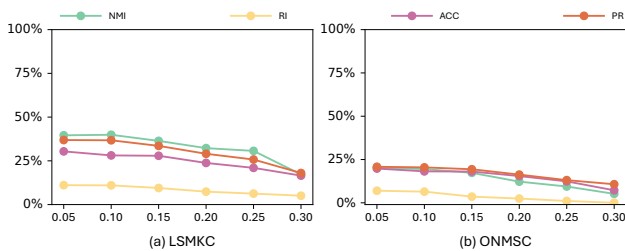

*Figure 5.* The impact of the proportion of perturbed samples on the BBCSport dataset.

*Figure 6.* The impact of the proportion of perturbed samples on the ProteidFold dataset.

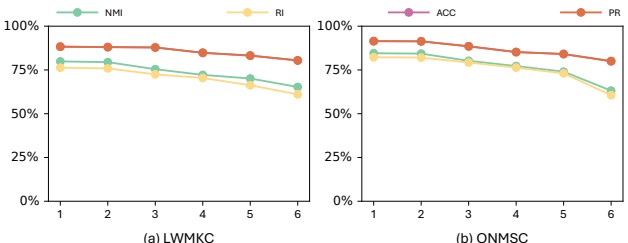

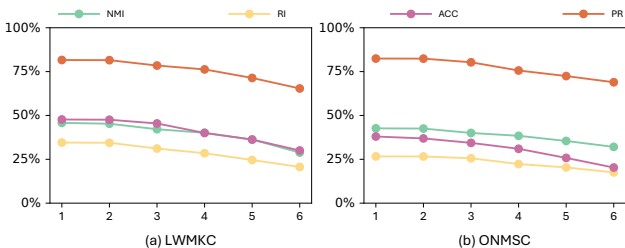

*Figure 7.* The impact of the number of perturbed views on the HW-6Views dataset.

*Figure 8.* The impact of the number of perturbed views on the Caltech101-7 dataset.

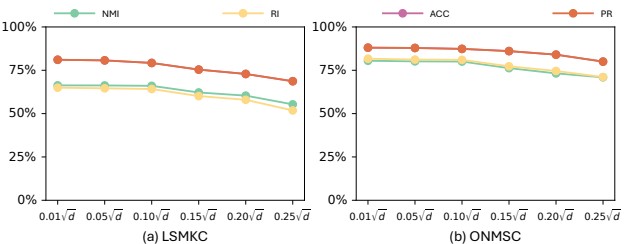

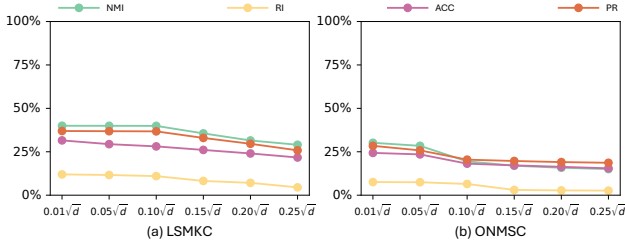

*Figure 9.* The impact of adversarial perturbation magnitude on the BBCSport dataset.

*Figure 10.* The impact of adversarial perturbation magnitude on the ProteinFold dataset.

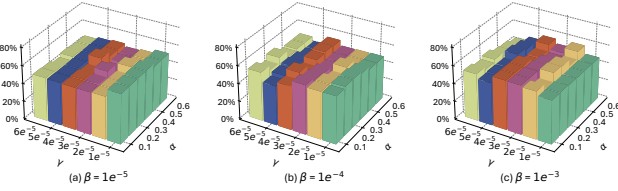



*Figure 11.* Effect of hyperparameters $\alpha$, $\beta$, and $\gamma$ on the MSRCv1 dataset, evaluated using the NMI metric.

*Figure 12.* Effect of hyperparameters $\alpha$, $\beta$, and $\gamma$ on the MSRCv1 dataset, evaluated using the ARI metric.

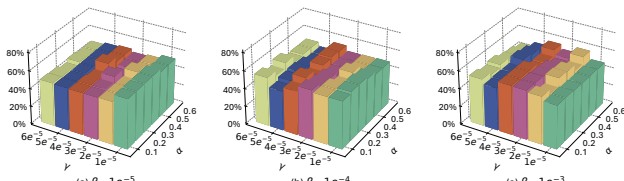

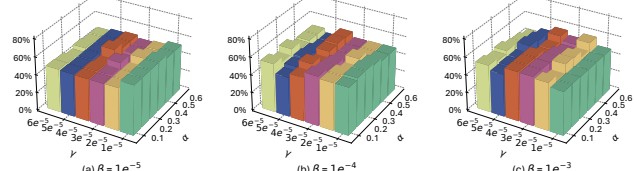

*Figure 13.* Effect of hyperparameters $\alpha$, $\beta$, and $\gamma$ on the MSRCv1 dataset, evaluated using the ACC metric.

*Figure 14.* Effect of hyperparameters $\alpha$, $\beta$, and $\gamma$ on the MSRCv1 dataset, evaluated using the PR metric.

the number of perturbed views $N_k^p$, and the magnitude of adversarial perturbations $\epsilon$. The results, shown in Figures 5–10, indicate that attack performance improves as these factors increase.

Additionally, we perform a sensitivity analysis on the three hyperparameters $\alpha$, $\beta$, and $\gamma$ in Eq. (16), which regulate the balance among loss terms. To ensure these loss terms have magnitudes comparable to the main terms, we select

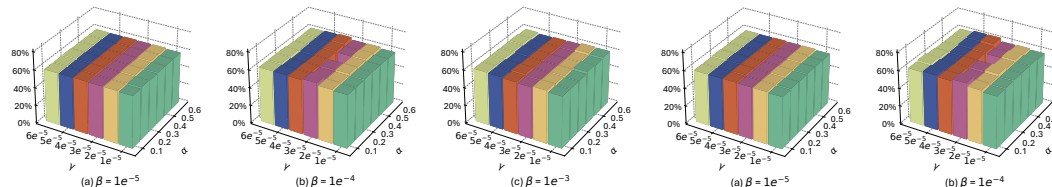

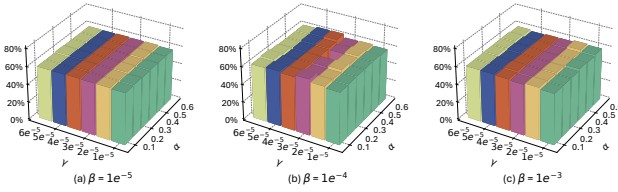

Figure 15. Effect of hyperparameters $\alpha$, $\beta$, and $\gamma$ on the BBCSport dataset, evaluated using the NMI metric.

Figure 16. Effect of hyperparameters $\alpha$, $\beta$, and $\gamma$ on the BBCSport dataset, evaluated using the ARI metric.

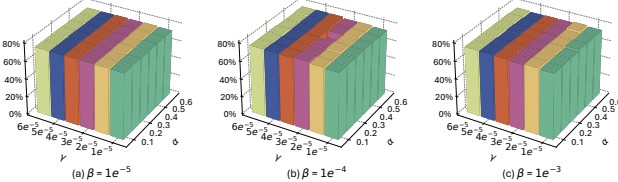

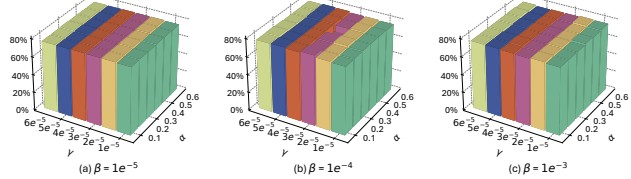

Figure 17. Effect of hyperparameters $\alpha$, $\beta$, and $\gamma$ on the BBCSport dataset, evaluated using the ACC metric.

Figure 18. Effect of hyperparameters $\alpha$, $\beta$, and $\gamma$ on the BBCSport dataset, evaluated using the PR metric.

Table 6. Comparison of time consumption for RAMKC, EAMKC, and AdvMKC on all six datasets (s).

| Dataset | Attack | SMKKM | EEOMVC | LSWMKC | LSMKC | MVC-LFA | LSMKKM | MKKM-SR |
|---|---|---|---|---|---|---|---|---|
| MSRCv1 | RAMKC | 1125.92 | 528.48 | 1476.00 | 151.00 | 142.68 | 169.28 | 43.00 |
| | EAMKC | 1162.00 | 605.64 | 1547.68 | 169.44 | 156.24 | 196.40 | 54.72 |
| | AdvMKC | 640.28 | 314.53 | 960.72 | 97.35 | 93.46 | 116.10 | 26.63 |
| BBCSport | RAMKC | 594.36 | 306.12 | 254.12 | 157.36 | 125.92 | 121.60 | 32.60 |
| | EAMKC | 626.55 | 345.18 | 329.95 | 188.18 | 157.55 | 153.64 | 43.25 |
| | AdvMKC | 42.26 | 187.33 | 185.83 | 89.77 | 80.81 | 75.95 | 27.87 |
| Protein Fold | RAMKC | 1158.16 | 1501.96 | 2737.96 | 615.16 | 397.04 | 508.36 | 136.64 |
| | EAMKC | 1356.02 | 1659.16 | 2830.79 | 729.16 | 419.07 | 623.62 | 168.40 |
| | AdvMKC | 193.34 | 915.09 | 335.79 | 391.49 | 252.52 | 323.06 | 94.13 |
| Caltech 101-7 | RAMKC | 5292.00 | 1033.56 | 7884.12 | 487.96 | 579.60 | 520.00 | 125.04 |
| | EAMKC | 5750.15 | 1149.57 | 8064.44 | 522.11 | 630.46 | 556.36 | 148.15 |
| | AdvMKC | 1092.53 | 672.34 | 1044.33 | 281.26 | 385.33 | 341.34 | 87.57 |
| Citeseer | RAMKC | 4161.24 | 1014.08 | 19228.40 | 769.24 | 423.64 | 1595.28 | 186.56 |
| | EAMKC | 4206.61 | 1048.15 | 2180.57 | 818.58 | 521.60 | 1683.57 | 190.21 |
| | AdvMKC | 640.36 | 645.45 | 1955.01 | 520.71 | 299.91 | 1168.52 | 137.82 |
| NUS-WIDE-SCENE | RAMKC | 5396.12 | 2187.44 | 36674.16 | 2651.68 | 975.56 | 2060.40 | 362.72 |
| | EAMKC | 5594.75 | 2302.42 | 39721.68 | 2721.95 | 994.56 | 2201.70 | 417.25 |
| | AdvMKC | 1068.54 | 1330.89 | 4167.51 | 1607.49 | 649.42 | 1588.76 | 272.86 |

$\alpha$ from $\{0.1, 0.2, 0.3, 0.4, 0.5, 0.6\}$, $\beta$ from $\{1e^{-5}, 1e^{-4}, 1e^{-3}\}$, and $\gamma$ from $\{1e^{-5}, 2e^{-5}, 3e^{-5}, 4e^{-5}, 5e^{-5}, 6e^{-5}\}$. The sensitivity analysis is conducted on the MSRCv1 and BBCSport datasets, with results shown in Figure 12–18. These results indicate that as long as the magnitudes of the loss terms remain balanced, variations in these parameters have minimal impact on attack performance.

**Time Consumption**. We evaluate the time consumption of AdvMKC in comparison to two baseline methods, using the same experimental settings as in Table 1. The results, presented in Table 6, demonstrate that AdvMKC requires less time while achieving the highest attack performance among all methods.

**Attack Efficiency**. Beyond attack performance, efficiency (Li et al., 2020; Liu et al., 2024; Yang et al., 2025a;b) is a crucial metric for evaluating adversarial attacks. Since the number of clustering operations directly affects computational cost, we assess the attack efficiency of AdvMKC in comparison to RAMKC and EAMKC.

To fully demonstrate the potential of AdvMKC, we set the number of selected samples to $N_d^p = 0.5N$, removing constraints on sample selection. Evaluations are conducted using SMKKM and ONMSC as victim MKC methods on the MSRCv1 and BBCSport datasets. The results, presented in Figures 19 - 22, show that AdvMKC consistently achieves optimal attack

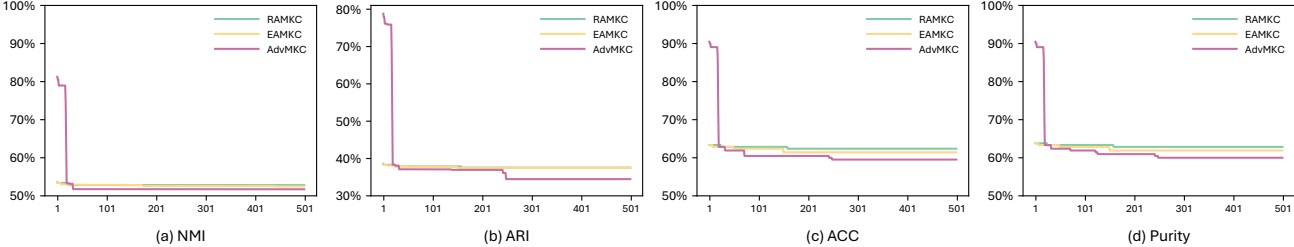

*Figure 19.* Attack efficiency of AdvMKC against the SMKKM algorithm on the MSRCv1 dataset. The horizontal axis represents the number of clustering operations performed.

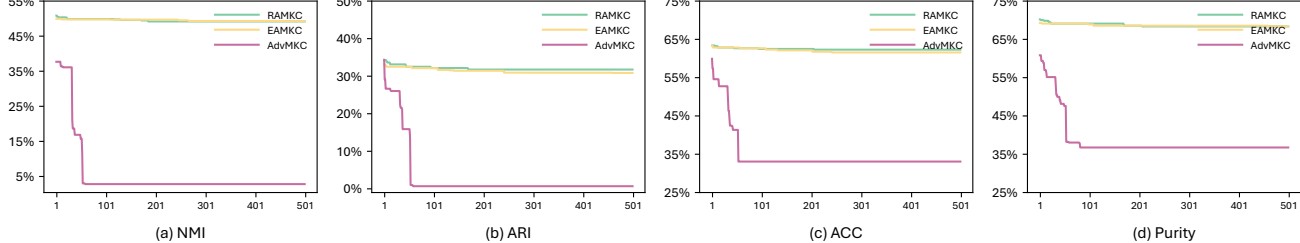

*Figure 20.* Attack efficiency of AdvMKC against the ONMSC algorithm on the MSRCv1 dataset. The horizontal axis represents the number of clustering operations performed.

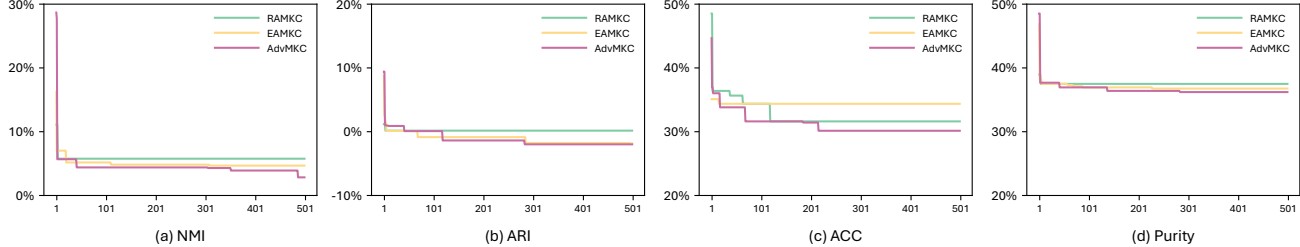

*Figure 21.* Attack efficiency of AdvMKC against the SMKKM algorithm on the BBCSport dataset. The horizontal axis represents the number of clustering operations performed.

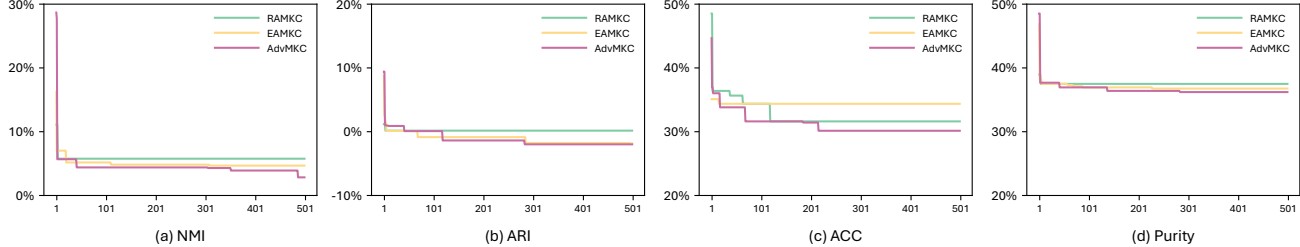

*Figure 22.* Attack efficiency of AdvMKC against the ONMSC algorithm on the BBCSport dataset. The horizontal axis represents the number of clustering operations performed.

performance with the fewest clustering operations. This efficiency is attributed to AdvMKC's ability to leverage the clusterer to approximate the behavior of the victim MKC methods.

It can be observed that AdvMKC does not attain the best performance during the initial stage of the attack. The performance difference stems from the distinct generation methods used in AdvMKC compared to the other two approaches. As shown in Appendix B, RAMKC injects Gaussian noise into the original data, while EAMKC optimizes the mean and variance of noise distributions using the LM-CMA strategy and a reward function. In contrast, AdvMKC employs a neural network to generate perturbations. Due to the initial parameter settings, the perturbation magnitude may be small at the beginning, limiting AdvMKC's performance in the early attack phase. However, once the memory buffer $\mathcal{B}$ is filled, the attacker can optimize the generator's parameters, resulting in improved attack performance.

