# OpenReview forum: "On the Adversarial Robustness of Multi-Kernel Clustering"
_ICML.cc/2025/Conference — ICML 2025 poster_

### Official Review · Reviewer_HijP · 2025-02-21

**Overall Recommendation:** 4

**Summary:**

This paper examines the vulnerability of MKC methods to adversarial perturbations—an area that remains understudied. The authors introduce AdvMKC, a reinforcement learning framework that generates subtle perturbations to deceive MKC methods in black-box settings. Using proximal policy optimization and an innovative generator-clusterer architecture, AdvMKC stabilizes clustering outcomes while reducing computational overhead. Extensive experiments across seven datasets and eleven MKC methods validate the framework's effectiveness, robustness, and transferability.

**Claims And Evidence:**

Yes. The experimental results strongly support the main claims in the submission. The results are presented clearly and thoroughly, with a comprehensive set of datasets demonstrating the method's effectiveness.

**Essential References Not Discussed:**

There are no essential references missing from the discussion.

**Experimental Designs Or Analyses:**

The experimental setup is generally appropriate, with a good selection of benchmarks and adversarial examples used to evaluate robustness.

**Methods And Evaluation Criteria:**

The proposed method leverages reinforcement learning—an approach particularly well-suited for generating adversarial perturbations.

**Other Comments Or Suggestions:**

The legends in the experimental results figures are too small and should be enlarged.

**Other Strengths And Weaknesses:**

Strengths
- This paper is pioneering in its examination of MKC methods' adversarial robustness under black-box settings.
- This paper proposes two additional comparison methods to fully demonstrate the effectiveness of the proposed approach.
- The motivation is clear and the paper is well-written and easy to follow.

Weaknesses
- In Table I, some cases show improved MKC method performance under adversarial conditions. The authors should clarify why this phenomenon occurs.
- In Eq. (16), the authors introduce two hyperparameters to balance three loss terms. However, they do not conduct sensitivity analysis in the evaluation section to demonstrate how these parameters impact the attack performance.
- Considering this is the first work to assess the adversarial robustness of MKC methods, the authors should release their source code to promote further research.

**Questions For Authors:**

- In Table I, some cases show improved performance of the MKC method under adversarial conditions. Could the authors clarify why this phenomenon occurs?
- In Eq. (16), the authors introduce two hyperparameters to balance three loss terms. Could the authors provide a sensitivity analysis in the evaluation section to demonstrate how these parameters impact the attack performance?

**Relation To Broader Scientific Literature:**

The paper makes significant contributions in the area of adversarial robustness in MKC, proposing a new reinforcement-learning-based adversarial attack framework (AdvMKC).

**Theoretical Claims:**

The proofs are generally well-structured and the key assumptions are clearly stated.

---

> ### Author Rebuttal · Authors · 2025-04-01
>
> **We sincerely appreciate Reviewer HijP’s thorough and constructive review. We provide point-by-point responses to the raised weaknesses as follows:**
>
> ---
>
> **W1:** In Table I, some cases show improved MKC method performance under adversarial conditions. The authors should clarify why this phenomenon occurs.
>
> **R1:** We appreciate the reviewer’s insightful comments. As noted in Subsection 6.1, to ensure the stealthiness of adversarial attacks, we introduce the hyperparameter $\epsilon$ to constrain the magnitude of injected noise. To ensure its effectiveness, $\epsilon$ is set to a very low value. However, these slight perturbations may lead the victim MKC method to learn more robust representations, resulting in a slight improvement in clustering performance. We will clarify this point in the revision.
>
> **W2:** In Eq. (16), the authors introduce two hyperparameters to balance three loss terms. However, they do not conduct sensitivity analysis in the evaluation section to demonstrate how these parameters impact the attack performance.
>
> **R2:** Thank you for the reviewer’s comments. Based on your suggestion, we conduct a sensitivity analysis on the hyperparameters $\alpha$, $\beta$, and $\gamma$ in Eq. (16), which regulate the balance among loss terms. To ensure comparable magnitudes across these terms, we select $\alpha$ from $\lbrace 0.1, 0.2, 0.3, 0.4, 0.5, 0.6\rbrace$, $\beta$ from $\lbrace e^{-5}, e^{-4}, e^{-3} \rbrace$, and $\gamma$ from $\lbrace e^{-5}, 2 e^{-5}, 3 e^{-5}, 4 e^{-5}, 5 e^{-5}, 6 e^{-5}\rbrace$. The analysis is conducted on the MSRCv1 and BBCSport datasets, with results shown in [the following eight figures](https://anonymous.4open.science/api/repo/AdvMKC-rebuttal-CC14/file/sensitivity_analysis.png?v=0fcf01a5). We conducted a total of **216 evaluations**. Due to the limitations of OpenReview.net, we present the evaluations on the following anonymous GitHub repository. For example, as shown in Figure 13, changes in $\alpha$, $\beta$, and $\gamma$ have no significant effect on the MNI metric for the BBCSport dataset. These results suggest that as long as the loss terms remain balanced in magnitude, variations in these parameters have minimal impact on attack performance.
>
> Experimental results are presented at https://anonymous.4open.science/r/AdvMKC-rebuttal-CC14/sensitivity_analysis.png.
>
> **W3:** Considering this is the first work to assess the adversarial robustness of MKC methods, the authors should release their source code to promote further research.
>
> **R3:** We appreciate the reviewer’s comments and acknowledge the concern. The code will be released upon the paper’s acceptance.

---

### Official Review · Reviewer_rAQk · 2025-03-05

**Overall Recommendation:** 3

**Summary:**

AdvMKC proposes a novel black-box adversarial attack for multi-kernel clustering that employs reinforcement learning—specifically, proximal policy optimization with an advantage function—within a generator-clusterer framework. This approach introduces minimal perturbations to mislead multi-kernel clustering while concurrently reducing computational overhead. Comprehensive theoretical analysis and experimental evaluations across several datasets and different MKC variants verify the method's efficacy, robustness, and transferability.

**Claims And Evidence:**

Yes, the submission’s claims are well-supported by both rigorous theoretical analysis and comprehensive experiments.Yes, the methods and benchmarks are well-suited to the problem.

**Essential References Not Discussed:**

None

**Experimental Designs Or Analyses:**

Yes

**Methods And Evaluation Criteria:**

Yes, the methods and benchmarks are well-suited to the problem.

**Other Comments Or Suggestions:**

None

**Other Strengths And Weaknesses:**

Strengths:
1. The study introduces the first black-box adversarial evaluation for multi-kernel clustering and presents an innovative reinforcement learning-based attack framework, AdvMKC.
2. The theoretical insights provided indicate that the impact of adversarial perturbations is governed by both their magnitude and frequency.
3. The proposed method has been rigorously validated on a diverse range of benchmark datasets.

Weaknesses:
1. Although the generator-clusterer framework is claimed to reduce computational costs, prior research has already enhanced multi-view clustering efficiency, and clustering processes typically do not entail significant runtime overhead. It would be beneficial for the authors to report the actual running times of each algorithm to substantiate this claim.
2. While the effects of parameters such as the selected sample count ($N_d^p$), perturbed view count ($N_k^p$), and perturbation magnitude ($\epsilon$) on attack performance are examined, the loss function in Equation (16), which comprises four terms and three hyper-parameters, lacks a comprehensive sensitivity analysis.
3. The occurrence of negative values in the ARI metric, as illustrated in Table 2, requires clarification on whether these outcomes are expected.

**Questions For Authors:**

Please refer to the weakness.

**Relation To Broader Scientific Literature:**

Yes. The paper extends established adversarial and clustering work by integrating reinforcement learning, perturbation strategies, and a generator-clusterer framework to robustify MKC.

**Theoretical Claims:**

Yes, the proofs for adversarial impact on MKC are verified.

---

> ### Author Rebuttal · Authors · 2025-04-01
>
> **We sincerely appreciate Reviewer rAQk's thorough and constructive review. We provide point-by-point responses to the raised weaknesses as follows:**
>
> ---
>
> **W1:** Although the generator-clusterer framework is claimed to reduce computational costs, prior research has already enhanced multi-view clustering efficiency, and clustering processes typically do not entail significant runtime overhead. It would be beneficial for the authors to report the actual running times of each algorithm to substantiate this claim.
>
> **R1:** We appreciate the reviewer’s insightful comments. We evaluate the time consumption of AdvMKC compared to two baseline methods under the same experimental settings as in Table 1. The results, presented in [the following table](https://anonymous.4open.science/api/repo/AdvMKC-rebuttal-CC14/file/running_time.png?v=12f17fd6), show that AdvMKC requires less time while achieving the highest attack performance among all methods. We conducted a total of **126 evaluations**. Due to the rebuttal's space limitations, we provide the evaluations via the following anonymous GitHub link. For example, when attacking the SMKKM MKC method on the MSRCv1 dataset, AdvMKC completes in 640.28 seconds, which is significantly lower than the 1259.92 seconds and 1162.00 seconds required by RAMKC and EAMKC, respectively. We will include these experimental results in the revision.
>
> Experimental results are presented at https://anonymous.4open.science/r/AdvMKC-rebuttal-CC14/running_time.png.
>
> **W2:** While the effects of parameters such as the selected sample count ($N_d^p$), perturbed view count ($N_k^p$), and perturbation magnitude ($\epsilon$) on attack performance are examined, the loss function in Equation (16), which comprises four terms and three hyper-parameters, lacks a comprehensive sensitivity analysis.
>
> **R2:** Thank you for the reviewer’s comments. Based on your suggestion, we conduct a sensitivity analysis on the hyperparameters $\alpha$, $\beta$, and $\gamma$ in Eq. (16), which regulate the balance among loss terms. To ensure comparable magnitudes across these terms, we select $\alpha$ from $\lbrace 0.1, 0.2, 0.3, 0.4, 0.5, 0.6 \rbrace$, $\beta$ from $\lbrace e^{-5}, e^{-4}, e^{-3} \rbrace$, and $\gamma$ from $\lbrace e^{-5}, 2 e^{-5}, 3 e^{-5}, 4 e^{-5}, 5 e^{-5}, 6 e^{-5} \rbrace$. The analysis is conducted on the MSRCv1 and BBCSport datasets, with results shown in [the following eight figures](https://anonymous.4open.science/api/repo/AdvMKC-rebuttal-CC14/file/sensitivity_analysis.png?v=0fcf01a5). We conducted a total of **216 evaluations**. Due to the limitations of OpenReview.net, we present the evaluations on the following anonymous GitHub repository. For example, as shown in Figure 13, changes in $\alpha$, $\beta$, and $\gamma$ have no significant effect on the MNI metric for the BBCSport dataset. These results suggest that as long as the loss terms remain balanced in magnitude, variations in these parameters have minimal impact on attack performance.
>
> Experimental results are presented at https://anonymous.4open.science/r/AdvMKC-rebuttal-CC14/sensitivity_analysis.png.
>
> **W3:** The occurrence of negative values in the ARI metric, as illustrated in Table 2, requires clarification on whether these outcomes are expected.
>
> **R3:** We appreciate the reviewer’s comments. The ARI can take negative values because it accounts for randomness in clustering assignments. It adjusts the Rand Index by considering the expected similarity between two random clusterings. If the observed clustering is worse than a random assignment, the ARI can be negative. We will include this clarification in the revision.

---

### Official Review · Reviewer_W7KP · 2025-03-06

**Overall Recommendation:** 3

**Summary:**

This manuscript addresses the underexplored vulnerability of MKC methods to adversarial perturbations. To evaluate the adversarial robustness of MKC in a black-box setting, the authors propose AdvMKC, a novel framework grounded in reinforcement learning. AdvMKC employs proximal policy optimization with an advantage function to ensure stable optimization and mitigate clustering instability. It introduces a generator-clusterer architecture, where the generator produces adversarial perturbations and the clusterer simulates MKC behavior, thereby reducing computational overhead. Experimental results demonstrate that AdvMKC is effective, robust, and transferable across various MKC techniques.

**Claims And Evidence:**

Yes, all claims presented in the manuscript are substantiated by experimental results and corresponding proofs.

**Essential References Not Discussed:**

No.

**Experimental Designs Or Analyses:**

I have reviewed the evaluation design and found that the authors conduct thorough assessments to evaluate the proposed method's effectiveness, robustness, and transferability.

**Methods And Evaluation Criteria:**

Yes, adversarial attacks on MKC methods have been insufficiently explored, and the proposed method is the first to address this issue. The datasets used in the evaluations are standard in MKC research.

**Other Comments Or Suggestions:**

In Figure 1, the font size of some text is too small. I recommend increasing the font size for better clarity.

**Other Strengths And Weaknesses:**

Strengths
1. This manuscript is the first to investigate the adversarial robustness of MKC methods in a black-box setting, where an attacker introduces imperceptible, targeted perturbations to deceive unknown MKC methods.
2. The manuscript provides valuable theoretical insights into the effects of adversarial perturbations on MKC methods.
3. The authors perform comprehensive evaluations across seven benchmark datasets and eleven MKC methods to assess the effectiveness, robustness, and transferability of the proposed method.

Weaknesses
1. The authors do not provide the code. If released, it would enhance reproducibility and be beneficial for the research community.
2. When evaluating the transferability of adversarial perturbations generated by the proposed method, the authors should clarify the meaning of the surrogate MKC method referenced in Table 3.
3. The authors should provide further details on the experimental setup, such as the specific view IDs used in adversarial attacks, given that each dataset comprises multiple data views.
4. In figures such as Figure 2 and Figure 3, the ACC metric curve is not clearly presented. However, in certain figures, such as Figure 10, it is clear that the four lines represent four distinct metrics. The authors should address this issue and provide clarification.

**Questions For Authors:**

Please consider addressing the points raised in the above weaknesses part.

**Relation To Broader Scientific Literature:**

This paper examines the adversarial robustness of MKC methods, an area that has not been previously studied.

**Theoretical Claims:**

I have reviewed the proof in the Appendix and found no issues.

---

> ### Author Rebuttal · Authors · 2025-04-01
>
> **We sincerely appreciate Reviewer W7KP’s thorough and constructive review. We provide point-by-point weaknesses to the raised questions as follows:**
>
> ---
>
> **W1:** The authors do not provide the code. If released, it would enhance reproducibility and be beneficial for the research community.
>
> **R1:** We appreciate the reviewer’s comments and acknowledge the concern. The code will be released upon the paper’s acceptance.
>
> **W2:** When evaluating the transferability of adversarial perturbations generated by the proposed method, the authors should clarify the meaning of the surrogate MKC method referenced in Table 3.
>
> **R2:** Thank you for the reviewer’s insightful comments. In Subsection 6.4, we consider a scenario where frequent queries to the victim MKC method are impractical. To address this, the attacker can utilize an alternative MKC method, referred to as the surrogate MKC method, to approximate the victim MKC method’s functionality. This allows the attacker to optimize the perturbation generator to produce adversarial noise. We will clarify the surrogate MKC method in the revision.
>
> **W3:** The authors should provide further details on the experimental setup, such as the specific view IDs used in adversarial attacks, given that each dataset comprises multiple data views.
>
> **R3:** We appreciate the reviewer’s comments. The evaluation details, including datasets, MKC methods, and compared methods, are provided in Appendix C. We conduct evaluations across seven datasets: MSRCv1, BBCSport, Caltech101-7, HW-6Views, Citeseer, NUS-WIDE-SCENE, and ProteinFold. To ensure a balanced evaluation, we select data views with low dimensionality and limit the total number of perturbed views to less than 50%. Specifically, for the MSRCv1 dataset, we perturb the first and fourth views, while for BBCSport and Citeseer, only the first view is perturbed. In the ProteinFold dataset, we perturb the first, second, third, and fourth views, whereas, in Caltech101-7, only the first and second views are affected. For HW-6Views, the perturbation is applied to the third and fourth views, and for NUS-WIDE-SCENE, the second and fifth views are perturbed. Additional details can be found in Table 4. We will further clarify this information and provide additional evaluation details in the revision.
>
> **W4:** In figures such as Figure 2 and Figure 3, the ACC metric curve is not clearly presented. However, in certain figures, such as Figure 10, it is clear that the four lines represent four distinct metrics. The authors should address this issue and provide clarification.
>
> **R4:** Thank you for the reviewer’s insightful comments. As shown in Tables 3 and 4, the accuracy and purity metrics coincide in some cases. Since clustering is an unsupervised task, both metrics rely on the best possible label assignment. The accuracy metric determines the optimal one-to-one mapping between cluster IDs and ground truth labels using the Hungarian algorithm [1], whereas the purity metric assigns each cluster to the most frequent ground truth label. When clusters closely align with single ground truth classes (i.e., each cluster predominantly contains data points from one class), both metrics yield identical values. In such cases, the best mapping in Hungarian accuracy and the dominant-class assignment in purity produce the same results. We will include further explanations in the revision.
>
> [1] Kuhn et al. The Hungarian Method for the Assignment Problem. 1955.

---

### Official Review · Reviewer_yajj · 2025-03-11

**Overall Recommendation:** 4

**Summary:**

The paper investigates the adversarial robustness of MKC in a black-box setting, a largely unexplored area. It introduces AdvMKC, a novel reinforcement-learning-based attack framework that injects imperceptible perturbations to mislead MKC methods. AdvMKC employs proximal policy optimization with an advantage function to handle clustering instability and utilizes a generator-clusterer framework to approximate MKC behavior while reducing computational costs. Theoretical analysis and extensive experiments on seven datasets and eleven MKC methods confirm the effectiveness, robustness, and transferability of AdvMKC.

**Claims And Evidence:**

The claims are generally supported by theoretical analysis and extensive experiments.

**Essential References Not Discussed:**

No

**Experimental Designs Or Analyses:**

Yes, the experiments on seven datasets and eleven MKC methods are reviewed.

**Methods And Evaluation Criteria:**

The proposed methods and evaluation criteria are well-aligned with the problem.

**Other Comments Or Suggestions:**

NA

**Other Strengths And Weaknesses:**

Strengths

•	This paper introduces AdvMKC, a reinforcement-learning-based attack with a perturbation generator and clusterer for efficient optimization.

•	This paper provides insights into the impact of adversarial perturbations on MKC performance.

•	This paper validates findings through experiments on seven datasets and eleven MKC methods, demonstrating effectiveness, robustness, and transferability.

Weaknesses

•	This paper assumes the victim MKC method operates as a black box with no direct access. Given this realistic constraint, where frequent queries are not feasible, how does the proposed attack function effectively in such a scenario?

•	Figures 11–14 show that as the number of clustering operations increases, AdvMKC outperforms the two baseline methods in attack effectiveness. However, in some cases, AdvMKC does not achieve the best performance during the initial attack phase. The authors should provide clarification on these observations.

**Questions For Authors:**

See above “Weakness”.

**Relation To Broader Scientific Literature:**

The paper extends adversarial robustness research to MKC, building on prior work in adversarial attacks and kernel methods.

**Theoretical Claims:**

Yes

---

> ### Author Rebuttal · Authors · 2025-04-01
>
> **We sincerely appreciate Reviewer yajj’s thorough and constructive review. We provide point-by-point responses to the raised weaknesses as follows:**
>
> ---
>
> **W1:** This paper assumes the victim MKC method operates as a black box with no direct access. Given this realistic constraint, where frequent queries are not feasible, how does the proposed attack function effectively in such a scenario?
>
> **R1:** We appreciate the reviewer’s insightful comments. Conducting attacks on multi-view clustering requires extensive queries to the victim MKC method, resulting in high resource consumption. To mitigate this, we propose two countermeasures:
>
> 1. We design a clusterer that emulates MKC behavior. During attacks, this clusterer provides rewards, eliminating the need to query the victim black-box MKC method. As demonstrated in Appendix D, AdvMKC achieves the highest attack performance with the same number of victim MKC queries.
> 2. When querying the victim MKC method is excessively costly, we propose an alternative approach: querying a surrogate MKC method under the attacker's control. As shown in Table 3, optimizing the generator using the surrogate MKC method enables the crafted perturbations to effectively mislead the victim MKC method.
>
> We will clarify this issue in the revision.
>
> **W2:** Figures 11–14 show that as the number of clustering operations increases, AdvMKC outperforms the two baseline methods in attack effectiveness. However, in some cases, AdvMKC does not achieve the best performance during the initial attack phase. The authors should provide clarification on these observations.
>
> **R2:** We appreciate the reviewer’s comments. The performance difference stems from the distinct generation methods used in AdvMKC compared to the other two approaches. As shown in Appendix C, RAMKC injects Gaussian noise into the original data, while EAMKC optimizes the mean and variance of noise distributions using the LM-CMA strategy and a reward function. In contrast, AdvMKC employs a neural network to generate perturbations. Due to the initial parameter settings, the perturbation magnitude may be small at the beginning, limiting AdvMKC’s performance in the early attack phase. However, once the memory buffer $\mathcal{B}$ is filled, the attacker can optimize the generator’s parameters, resulting in improved attack performance. We will address this concern in the revision.

---

### Decision · Program_Chairs · 2025-05-01

**Decision:**

Accept (poster)

**Comment:**

Thank you for submitting your work to ICML 2025, and for your efforts in the rebuttal phase to clarify the reviewers' concerns. The authors should provide more comprehensive details about the experimental setup and enhance the quality and clarity of the figures in the camera-ready version. Since all the reviewers affirmed the novelty of the authors' work, I therefore recommend acceptance of the paper.